

# Tensorization of neural networks for improved privacy and interpretability

José Ramón Pareja Monturiol[1,2*], Alejandro Pozas-Kerstjens[3] and David Pérez-García[1,2]

**1** Departamento de Análisis Matemático, Universidad Complutense de Madrid, 28040 Madrid, Spain
**2** Instituto de Ciencias Matemáticas (CSIC-UAM-UC3M-UCM), 28049 Madrid, Spain
**3** Group of Applied Physics, University of Geneva, 1211 Geneva, Switzerland

* joserapa@ucm.es

## Abstract

We present a tensorization algorithm for constructing tensor train/matrix product state (MPS) representations of functions, drawing on sketching and cross interpolation ideas. The method only requires black-box access to the target function and a small set of sample points defining the domain of interest. Thus, it is particularly well-suited for machine learning models, where the domain of interest is naturally defined by the training dataset. We show that this approach can be used to enhance the privacy and interpretability of neural network models. Specifically, we apply our decomposition to (i) obfuscate neural networks whose parameters encode patterns tied to the training data distribution, and (ii) estimate topological phases of matter that are easily accessible from the MPS representation. Additionally, we show that this tensorization can serve as an efficient initialization method for optimizing MPS in general settings, and that, for model compression, our algorithm achieves a superior trade-off between memory and time complexity compared to conventional tensorization methods of neural networks.

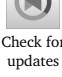
# 1  Introduction

Neural networks (NNs) have become highly effective tools for a variety of machine learning tasks, including regression, classification, and modeling probability distributions. This success stems in part from the flexibility they provide in designing architectures tailored to the specifics of each problem, as well as advances in optimization techniques and significant increases in computational power. A prominent example of this progress is the rapid evolution of deep convolutional networks—from early models like AlexNet [1] to more complex architectures

like VGGNet [2], GoogleNet [3] and ResNet [4]—which, powered by specialized hardware, have enabled efficient representation of increasingly complex data. Recently, transformer-based architectures have pushed this evolution further, with Large Language Models [5–8] now capable of modeling high-dimensional, intricate distributions across diverse data sources, including text, images, and audio.

This versatility has expanded NNs into fields such as quantum many-body physics, where their expressive power enables them to approximate quantum states within exponentially large Hilbert spaces, thereby mitigating the *curse of dimensionality*. This capability has supported applications such as quantum state tomography [9–12], ground-state wave function approx-imation [13–17], and phase classification of quantum systems [18–20]. A variety of archi-tectures has been utilized in these applications, including Restricted Boltzmann Machines (RBMs) [13, 15], Convolutional Neural Networks (CNN) [11], as well as Recurrent Neural Networks [16], and transformer-based models [17], inspired by the progress in language mod-eling.

Despite their successes, NNs have a significant drawback that limits their use in many ap-plications: they function as *black boxes*, with no direct means of understanding their inner workings. Interpreting the output of a NN typically requires *post hoc* techniques to explain its decisions [21, 22]. An alternative approach is to develop models that are inherently in-terpretable [23, 24]. However, in practice, the improvements in interpretability tend to be associated to limitations in the expressive capacity.

Another significant concern with NNs is privacy, especially with the advent of generative models trained on vast datasets from the internet [25, 26]. Beyond copyright issues, the lack of interpretability makes it difficult to determine whether sensitive information has been memo-rized. Current techniques aim to mitigate this risk by training models through reinforcement learning with human feedback, thereby reducing the likelihood of generating private data [27]. However, recent research reveals that these barriers can be overcome to extract training data, potentially compromising privacy [28, 29]. Additionally, gradient-based optimization tech-niques have been shown to cause data leakage, as information from the training dataset can create identifiable patterns within the NN parameters [30].

Among the possible solutions to these issues are Tensor Network (TN) models. Tensor net-works provide efficient representations of high-dimensional tensors with low-rank structure. Originating in condensed matter theory, they were developed as tractable representations of quantum many-body states, facilitating the study of entanglement, symmetries, and phase transitions [31–33]. Due to their practical efficiency, TNs have become a fundamental tool to simulate large quantum systems [34–39].

Recently, TNs have been proposed as quantum-inspired machine learning models for broader tasks. Initial studies in this area [40, 41] implemented 1D models known as Matrix Product States (MPS) [42] or Tensor Trains (TTs) [43], the latter being the customary term in this context, which we will adopt throughout this work. Subsequent research introduced the use of more complex tensor networks, such as Trees [44, 45], MERAs [46] or PEPS [47], for both supervised and unsupervised learning tasks.

Typically, TN models are trained through gradient descent methods, similarly to NNs. How-ever, one drawback of these algorithms is the number of hyperparameters that must be adjusted for effective optimization. TNs present additional challenges, as it is necessary to determine how to fix or update the inner ranks of the tensors composing the network. Additionally, the application of a TN involves the contraction of multiple tensors, making computations highly sensitive to small perturbations, which can cause rapid explosion or vanishing of outputs and gradients. This sensitivity complicates the design of effective initialization methods. Although some initialization strategies have been proposed [48, 49], they are often not generalizable across different network structures or data embeddings.

Despite these issues, TN models offer certain advantages over NNs. For instance, TNs have been shown to outperform other deep and classical algorithms on tabular data in anomaly detection tasks [50]. Furthermore, from their successful application in representing complex quantum systems, TNs are known to be effective in modeling probability distributions, especially when the network layout aligns with the data structure (e.g., using 2D PEPS for image modeling [47]). This structural alignment enables TNs to access critical information directly, such as the correlations within the data (i.e., the entanglement structure when studying quantum systems), or efficient computation of marginal and conditional distributions. This interpretability has been highlighted in recent studies [51, 52]. The capacity to model entire distributions and compute conditional probabilities has been leveraged to create TN models capable of both classification and generation, suggesting potential robustness against adversarial attacks [53, 54]. Regarding the previously mentioned data leakages that arise as a side effect of gradient descent training, TNs have been proposed as a solution due to the explicit characterization of their gauge freedom [30]. This property allows for the definition of new sets of parameters without altering the behavior of the black-box, thereby eliminating potential patterns introduced by training data.

Another key aspect that has drawn attention to TNs is their efficiency. This feature has already been utilized to compress pre-trained NNs by converting each linear layer in the model into a TN using low-rank tensor decompositions [55–62]. While this approach reduces the number of parameters, enabling complex models to fit on smaller devices, it does not always decrease the computational cost of applying the model unless very small ranks are selected. Furthermore, this layer-wise tensorization does not contribute to making models more interpretable or private, as a single TN model would.

Motivated by this, we aim to employ low-rank tensor decompositions to transform the entire NN black-box into a single TN, thereby enhancing interpretability and privacy in the reconstructed model. To achieve this, we introduce an adaptation of the sketching approach proposed by Hur *et al.* [63] for approximating probability distributions with TTs. This algorithm, Tensor Train via Recursive Sketching from Samples (TT-RSS), is better suited for decomposing NNs (those modeling probability distributions) when provided with a set of training samples that define the subregion of interest within the model's domain. Using this method, we achieve promising results in approximating image (Bars and Stripes, MNIST [64]) and speech (CommonVoice [65]) classification models, as well as in approximating more general functions beyond probability distributions.

## 1.1 Contributions

- We propose a heuristic that reduces the complexity of sketching for decomposing functions into TT format. Our algorithm is well-suited for scenarios of high dimensionality and sparsity, such as NN models, while also being applicable to general functions to accelerate computations. An implementation of our method is available in the open-source Python package TensorKrowch [66].

- Using the proposed algorithm, we extend the privacy experiments conducted in Ref. [30] to more realistic scenarios. Specifically, we demonstrate that the parameters of NN models trained to classify voices by gender are significantly influenced by the predominant accent present in the dataset. By tensorizing and re-training, we obtain TN models with comparable accuracies to the NNs, while substantially mitigating data leakage.

- As a demonstration of the interpretability capacities of TNs, we apply our decomposition to tackle a relevant problem in condensed matter physics. Namely, we reconstruct the exact AKLT state [67] from black-box access to the amplitudes of a limited number of spin

configurations. From the reconstruction, we compute an order parameter for symmetry-protected topological phase estimation, which is accessible through the local description of the tensors constituting the TN state [68]. We propose that this methodology, when extended to higher-dimensional cases, could provide deeper insights into the properties of systems represented by neural network quantum states (NNQS) [15].

- Finally, we discuss how this tensorization method can serve as a general approach to randomly initialize TTs, supporting various embeddings and sizes. Additionally, we argue that decomposing NNs in this manner can effectively compress models, achieving a better trade-off between memory and time efficiency compared to previous tensorization methods.

## 1.2 Organization

The remaining sections of the Introduction outline the main topics of this work, including a brief literature review of prior tensorization methods (Section 1.3) and relevant work on privacy and interpretability (Section 1.4). Additionally, the notations used throughout the paper are detailed in Section 1.5.

In Section 2, we introduce the proposed tensorization algorithm, TT-RSS, with a step-by-step explanation provided in Section 2.2. Its continuous adaptation is discussed in Section 2.3, while its computational complexity is analyzed in Section 2.4. Additionally, we discuss the close relation between the sketching approach utilized and the Tensor Train via Cross Interpolation algorithm [69] in Section 2.5.

To evaluate the algorithm's performance in terms of both time and accuracy, we present a series of experiments in Section 3 across various scenarios, including general non-density functions (Section 3.1) and NN models (Section 3.2). The impact of different hyperparameters on the algorithm's performance is analyzed in Section 3.3.

The primary contributions of this work are presented in Section 4. These include an experimental demonstration of a privacy attack on NN models and the proposed defense using TN models (Section 1.4), as well as the application of TT-RSS to find a TT representation of the AKLT state, enabling estimation of the symmetry-protected topological phase (Section 4.2).

Additional results are provided in Section 5, highlighting the use of tensorization as a general initialization method for TTs (Section 5.1) and demonstrating that representing a model as a single TN offers a better memory-time efficiency trade-off compared to conventional NN tensorization techniques (Section 5.2).

Finally, conclusions are presented in Section 6.

## 1.3 Prior work on tensorization

Our primary objective is to decompose NN models, which approximate probability distributions, into a TT. Specifically, we consider a non-negative, $n$-dimensional function $\hat{p}_{NN}$ (the neural network) that approximates an underlying density $p$, and aim to construct a new approximation of $p$, $\hat{p}_{TT}$ (in TT form), using a limited number of evaluations of $\hat{p}_{NN}$. For the decomposition of high-dimensional tensors into TT form, several algorithms have demonstrated promising results. Two primary techniques have been introduced to avoid more sophisticated and computationally expensive methods, such as Singular Value Decomposition (SVD), which can be problematic in high-dimensional settings. These techniques are based on cross interpolation and random sketching.

### 1.3.1 Cross interpolation

Unlike the SVD, which identifies a set of vectors spanning the column and row spaces inherently, cross interpolation [70], also known as pseudoskeleton [71] or CUR decomposition [72], reconstructs a matrix using $r$ selected columns and $r$ selected rows. For a rank-$r$ matrix, this decomposition can be exact; for matrices of higher rank, the error is generally larger compared to SVD, but the method is significantly faster. Building on this idea, it was shown that iterative application of cross interpolation can decompose a high-dimensional tensor into a TT format [73]. To minimize the decomposition error, an additional optimization step—namely, the `maxvol` routine—is required to identify the $r$ columns and $r$ rows whose intersection forms the submatrix with maximum volume, a criterion proven to yield the lowest error [74]. As this combinatorial optimization is NP-hard, greedy algorithms have been developed to accelerate the process while maintaining practical performance [75]. This approach is commonly referred to as Tensor Train via Cross Interpolation (TT-CI).

In earlier implementations, cross interpolation was applied directly to the entire tensor, which was treated as a matrix. This approach becomes computationally infeasible for high-dimensional tensors due to the exponential growth of their *matricizations*. To overcome this limitation, a parallel version of TT-CI specifically designed for high-dimensional scenarios was introduced in Ref. [69]. This method begins with an initial guess for the rows and columns, referred to as *pivots*, for all matricizations of the tensor. The initial guess forms a preliminary TT approximation, which is subsequently refined through iterative sweeps along the TT chain to optimize the pivots at each split. Additionally, the method can incorporate rank-revealing decompositions, such as SVD, as subroutines to determine the precise ranks needed to achieve a desired level of accuracy.

Although originally designed for tensor decomposition, TT-CI can also be applied to continuous functions by *quantizing* them, that is, discretizing the function's domain to represent it as a tensor [76]. Promising results using this approach were demonstrated in subsequent work [77], where TT-CI was applied across a broad range of tasks. This study emphasized the importance of selecting an effective initial set of pivots, as poor choices could limit exploration of the function's entire domain. Further advances include extensions to Tree TNs for hierarchical data representation [78].

### 1.3.2 Sketching

As noted, the goal of cross interpolation is to identify a set of vectors sufficient to span the column (or row) space of a matrix. An alternative approach is to use random projections, where random combinations of columns (or rows) are computed to generate random vectors in the range of the matrix. This idea is inspired in part by the Johnson-Lindenstrauss lemma [79], which states that a set of $l$ points in a high-dimensional Euclidean space can be embedded into an $m$-dimensional random subspace, where $m$ is logarithmic in $l$ and independent of the ambient dimension, while preserving pairwise distances within an arbitrarily small multiplicative factor. While various strategies can be used to generate these projections, the most general approach involves random Gaussian (or orthogonal) projections. Such random projections, often referred to as *sketches*, have been employed to accelerate numerous linear algebra routines, including the SVD [80–82].

Similar to cross interpolation, some algorithms based on random sketching have been extended to decompose tensors into TT form. Shi *et al.* [83] proposed a novel scheme where, instead of performing multiple sweeps along the TT chain to iteratively optimize tensors, a single sweep suffices. In this method, each tensor is computed by solving an equation formulated using the column spaces of subsequent matricizations of the tensor, which are derived from random sketches. However, for a $k \times l$ matrix $A$, multiplying by a random projection matrix

$\Omega \in \mathbb{R}^{l \times m}$ requires $O(klm)$ operations. In high-dimensional scenarios where $A$ represents the matricization of an $n$-dimensional tensor, $l$ scales exponentially with $n$, rendering matrix multiplication computationally infeasible in terms of both storage and time. To mitigate storage costs, random projections can be constructed using the Khatri-Rao product [84], although this approach does not reduce time complexity.

In subsequent work, Hur *et al.* [63] applied a similar approach to decompose an empirical distribution $\hat{p}_{\mathrm{E}}$, aiming to derive a TT approximation of the underlying density $p$. For efficiency, their algorithm assumes that the number of samples $N$ used to construct the empirical distribution is not excessively large, as the computation of random projections scales linearly with $N$. However, achieving a good approximation of $p$ requires $N$ to grow exponentially with the dimension $n$, making this method impractical for high-dimensional scenarios. The algorithm, Tensor Train via Recursive Sketching (TT-RS), has the notable advantage of being natively applicable to continuous functions by utilizing continuous embeddings, thereby avoiding the need for explicit quantization.

This framework has seen further advancements, such as the introduction of kernel density estimation to smooth the empirical distribution, as demonstrated in Ref. [85]. Additionally, extensions to hierarchical tensor structures have been developed [86, 87].

The approach proposed by Hur *et al.* [63] emphasizes not finding the most accurate approximation to $\hat{p}_{\mathrm{E}}$ but rather constructing a different approximation of $p$ in TT form. In our work, we follow this scheme, where the initial approximation $\hat{p}_{\mathrm{NN}}$ is provided by a NN, rather than by an empirical distribution. Consequently, we favor TT-RS over TT-CI for three primary reasons: (i) random sketches help reduce the variance of the approximation $\hat{p}$, (ii) SVD can be performed in each step without requiring modifications to the algorithm, and (iii) the method's applicability to continuous functions without increasing the number of tensors in the TT is crucial for maintaining efficiency. This is particularly important as we aim to avoid creating TT models with significantly more parameters than the original NNs, which could occur if we discretize the network to very high precision. Although such continuous adaptation is also feasible in TT-CI, TT-RS provides a more favorable approach. It allows, in a single sweep, to obtain the discrete tensors corresponding to the specific continuous embeddings, while simultaneously performing SVD to determine the appropriate ranks.

However, as discussed earlier, TT-RS with random sketches is generally inefficient, especially in the case we consider where the approximation $\hat{p}_{\mathrm{NN}}$ is not an empirical distribution. To address this, we incorporate the core idea of cross interpolation into TT-RS: first selecting a set of columns and rows to reconstruct the entire tensor. Unlike cross interpolation, where a rank-$r$ matrix is reconstructed from $r$ columns and $r$ rows with a submatrix of maximum volume at their intersection, we select a larger number $N > r$ of columns and rows, which may not be optimal in the sense of maximum volume. A random projection is then applied to this subspace to reduce variance, followed by SVD to determine the proper rank $r$.

In our case studies, we assume that the selected $N$ columns and rows correspond to points in the function's domain where the function exhibits significant behavior. For instance, when decomposing NNs, these points can be elements of the training dataset or samples drawn from the same distribution. Since these points are the ones the model was trained on, they represent the subregion of interest that the TT must capture, avoiding flat or noisy regions of the density. In scenarios where training points are unavailable, Monte Carlo techniques could be used to sample points with high probability. It should be noted that the selection of these points is heuristic, and we do not prove its optimality in this work.

## 1.4 Prior work on applications

Although we present a detailed description of the TT-RSS algorithm, which could be applicable to general functions, the primary contributions of this work are centered on its application to

NNs, highlighting the potential advantages that TN models have over them. Specifically, we argue that obtaining models as a single TN is beneficial in terms of privacy and interpretability.

### 1.4.1 Privacy

Preserving the privacy of users whose data is utilized in data analysis or machine learning processes is centered on safeguarding personal information that is sensitive or unintended for public exposure. Unlike security, which focuses on preventing unauthorized access to data, privacy aims to prevent the inference of personal information not only from access to the data but also from the processes where the data was employed.

Given the diversity of privacy-related attacks, a variety of concepts and defense mechanisms have been developed to address specific scenarios. Among these, Differential Privacy (DP) [88] stands out as a widely adopted framework due to its rigorous approach to assessing privacy risks and designing mitigation strategies. DP measures the likelihood that an attacker can determine whether a specific user's data was included in a statistical process. Specifically, a randomized algorithm $\mathcal{A}$ is $\varepsilon$-DP if, for any set of possible outcomes $\mathcal{S}$ in the range of $\mathcal{A}$, the following condition holds:

$$\log\left(\frac{\mathrm{P}[\mathcal{A}(D) \in \mathcal{S}]}{\mathrm{P}[\mathcal{A}(D') \in \mathcal{S}]}\right) \le \varepsilon\,, \tag{1}$$

where $D$ and $D'$ are datasets differing by a single element. This metric, akin to the Kullback-Leibler divergence, quantifies privacy protection through the parameter $\varepsilon$. It enables the design of random mechanisms by introducing a precisely calibrated amount of noise to achieve a desired privacy level $\varepsilon$, based on the *sensitivity* of the function being protected [89,90].

Several methods have been proposed to train Deep Neural Networks (DNNs) with DP. These include adding noise to input data [91] or to gradients during each step of gradient descent [92]. The latter, known as DP-SGD, has gained prominence as it ensures DP at the training level, making it independent of data acquisition or model deployment processes. However, DP-SGD is computationally expensive and negatively impacts model performance, including accuracy and fairness [93]. To address these challenges, Ref. [94] adapted DP-SGD to a training scheme involving low-rank updates to weight matrices, as in Ref. [95]. Additionally, pruning techniques have been proposed as an alternative to achieve DP without significantly affecting the training process [96,97].

Nevertheless, the practical application of DP faces certain limitations. For instance, while the framework provides theoretical bounds on privacy loss, its practical implications are often unclear, requiring reliance on experimental evaluation [98]. Moreover, DP primarily aims to protect against *membership inference attacks*, which reveal the presence of specific data points in a dataset [99]. However, it offers limited protection against *property inference attacks*, which aim to extract properties of the entire dataset. In this regard, Ref. [30] identified a novel form of data leakage arising from training with gradient descent methods. This phenomenon is experimentally examined in Section 4.1, demonstrating that NN models trained to classify voices by gender exhibit parameters that are distinguishable based on the predominant accent in the training dataset.

Additional strategies have been explored to enhance privacy in production-level models while minimizing the impact on accuracy. One example is training models with reinforcement learning with human feedback to reduce the memorization of sensitive information [27]. However, studies have shown that such approaches remain vulnerable to exploitation, raising significant privacy concerns [28,29].

### 1.4.2 Interpretability

Tensor networks have been acclaimed in the field of condensed matter physics due to their interpretability, which allows to easily study properties of quantum systems. In particular, TN representations of quantum states have been pivotal in studying topological phases of matter by capturing how global symmetries affect the local tensors constituting the network.

Specifically, it can be seen that two states are equal if there is a gauge transformation and a phase that transforms one representation to the other [42, 100]. This implies that, for translationally invariant (TI) TTs, under the action of global on-site, reflection, or time-reversal symmetries, the local tensors building up the TT must transform trivially up to a phase. This leads to writing TT tensors in terms of irreducible representations of a symmetry group $G$ [101]. These irreducible representations on the virtual degrees of freedom form non-linear projective representations [102–105].

Further work showed that injective, TI TTs [42] belong to the same phase if they can be interpolated without topological obstructions. However, when symmetry is imposed, the interpolation path is constrained and topological obstructions might occur, leading to phase transitions at the points where the interpolating TT becomes non-injective [103, 104]. These phases, known as Symmetry-Protected Topological (SPT) phases, are classified by the second cohomology group for gapped quantum spin systems [106, 107].

Unlike symmetry-breaking phases, which can be distinguished by local order parameters, SPT phases require non-local observables, like string order parameters, for detection [108]. A key example is the AKLT state, which exhibits non-trivial projective symmetry representations protected by multiple symmetries, including on-site $\mathbb{Z}_2 \times \mathbb{Z}_2$ symmetry [67]. Reference [68] introduced an order parameter to estimate the SPT phase of the AKLT state, which can be computed from the TI TT representation of the state. This approach is followed in Section 4.2.1 to estimate the AKLT SPT phase from the TT representation obtained via TT-RSS. Tensor networks, as such, remain invaluable due to their capacity to encode both the entanglement structure and symmetry properties of quantum states. For a more comprehensive exploration of this field, including its extensions to higher dimensions, we refer readers to the review by Cirac *et al.* [33] and the references therein.

Beyond phase classification, the interpretability capacities of TNs extend to other areas. Loop-less networks like TTs and Trees offer direct and efficient access to critical information such as marginal or conditional distributions, mutual information, and entropy, which has found applications in explainable machine learning [51, 52].

### 1.5 Notations

In this work we are interested in decomposing $n$-dimensional continuous functions $f : X_1 \times \cdots \times X_n \to \mathbb{R}$, with $X_1, \ldots, X_n \subset \mathbb{R}$. Although our primary goal is to tensorize densities, the methodology can be applied to general functions, and thus we do not assume $f$ is non-negative or normalized. Note that $f$ represents a general function used in the algorithm's description, while $p$ refers to densities, and $\hat{p}$ denotes approximations of $p$. These may serve as specific instances of $f$.

We will generally treat discrete functions $A : [d_1] \times \cdots \times [d_n] \to \mathbb{R}$, where $[n]$ denotes $\{1, \ldots, n\}$ for any $n \in \mathbb{N}$, as high-order tensors. To contract tensors, we adopt the Einstein convention, which assumes that a summation occurs along the common indices of the tensors. For instance, for matrices $A$ and $B$ of sizes $m \times n$ and $n \times k$, respectively, the contraction of the second index of $A$ with the first index of $B$ is expressed as:

$$A(i, j)B(j, k) := \sum_{j=1}^{n} A(i, j)B(j, k).$$
(2)

Furthermore, if we wish to restrict a summation to a specific set of values $J$ for the index $j$, we denote it as $A(i, J)$, and the corresponding sum along indices in $J$ as:

$$A(i, J)B(J, k) := \sum_{j \in J} A(i, j)B(j, k). \tag{3}$$

This notation will also be extended to continuous functions, replacing summations with integrals where necessary.

Typically, functions are restricted to a subset defined by a collection of selected points, referred to as *sketch samples* or, more commonly, *pivots*, denoted as $\mathbf{x} = \{(\mathbf{x}_1^i, \ldots, \mathbf{x}_n^i) : \mathbf{x}_j^i \in X_j\}_{i=1}^N$. The set of possible values for the $k$-th variable in $\mathbf{x}$ is represented as $\mathbf{x}_k = \{\mathbf{x}_k^i\}_{i=1}^N$. For integers $a < b$, the notation $a : b$ represents the set $\{a, a+1, \ldots, b\}$, which is useful for describing subsets of variables. For example, $\mathbf{x}_{a:b} = \{(\mathbf{x}_a^i, \ldots, \mathbf{x}_b^i)\}_{i=1}^N$. We use $\mathbf{x}^i = (\mathbf{x}_1^i, \ldots, \mathbf{x}_n^i)$ to select a single point, and subscript notation may also be applied to specify subsets of variables.

Although all pivots are assumed to be distinct, some may share the same values for certain variables. As a result, the sets $\mathbf{x}_k$ and $\mathbf{x}_{a:b}$ may contain fewer than $N$ unique elements. To account for this, we extend the subscript notation to specify the number of unique elements in each set. Specifically, we denote that there are $N_k$ unique elements in $\mathbf{x}_k$ and $N_{a:b}$ unique elements in $\mathbf{x}_{a:b}$.

This notation is further extended to group sets of variables in order to treat functions as matrices, commonly referred to as the *matricizations* or *unfolding matrices* of a tensor. For a function $f(x_1, \ldots, x_n)$, its $k$-th unfolding matrix is denoted as $f((x_1, \ldots, x_k), (x_{k+1}, \ldots, x_n))$ or, more concisely, $f(x_{1:k}, x_{k+1:n})$. This notation can be generalized to higher-order tensors by grouping variables into more sets, for instance, $f((x_1, \ldots, x_{k-1}), x_k, (x_{k+1}, \ldots, x_n)) = f(x_{1:k-1}, x_k, x_{k+1:n})$. When clear from the context, the explicit formation of a matricization may be omitted, and elements such as $f(x_{1:k-1}, x_k, x_{k+1:n})$ or $f(x_{1:k}, x_{k+1:n})$ will be treated interchangeably.

Finally, we define the tensor train format:

**Definition 1.1.** A function $f$ admits a tensor train representation with ranks $r_1, \ldots, r_{n-1}$ if there exist functions $G_1 : X_1 \times [r_1] \to \mathbb{R}$, $G_k : [r_{k-1}] \times X_k \times [r_k] \to \mathbb{R}$ for all $k \in \{2, \ldots, n-1\}$, and $G_n : [r_{n-1}] \times X_n \to \mathbb{R}$, referred to as *cores*, such that

$$f(x_1, \ldots, x_n) = G_1(x_1, \alpha_1)G_2(\alpha_1, x_2, \alpha_2) \cdots G_{n-1}(\alpha_{n-2}, x_{n-1}, \alpha_{n-1})G_n(\alpha_{n-1}, x_n), \tag{4}$$

for all $(x_1, \ldots, x_n) \in X_1 \times \cdots \times X_n$. Note that the right-hand side of the expression above has, per Einstein's convention, implicit sums over the indices $\{\alpha_1, \ldots, \alpha_{n-1}\}$.

When the function $f$ is discrete, the cores are discrete tensors. Otherwise, the cores are continuous functions, expressed as linear combinations of a small set of *embedding functions*. This is, given a set of $d_k$ one-dimensional functions $\{\phi_k^{i_k}\}_{i_k=1}^{d_k}$, we decompose the cores as

$$G_k(\alpha_{k-1}, x_k, \alpha_k) = \breve{G}_k(\alpha_{k-1}, i_k, \alpha_k)\phi_k(i_k, x_k), \tag{5}$$

where $\phi_k(i_k, x_k) = \phi_k^{i_k}(x_k)$.

## 2 Description of the algorithm

In this section we provide a detailed description of our proposed algorithm, Tensor Train via Recursive Sketching from Samples, or TT-RSS, presented in a similar form to the presentation of TT-RS in Ref. [63]. We first consider the simplified case of a discrete function $f : [d_1] \times \cdots \times [d_n] \to \mathbb{R}$ and assume it admits a TT representation with ranks $r_1, \ldots, r_{n-1}$.

Additionally, we are given a set of $N$ pivots, $\mathbf{x} = \{(\mathbf{x}_1^i, \dots, \mathbf{x}_n^i) : \mathbf{x}_j^i \in [d_j]\}_{i=1}^N$, which, as stated previously, could represent training points from the model or points defining a region of interest within the domain of the function $f$.

## 2.1 Main idea

The algorithm is based on the observation that $f$ can be treated as a low-rank matrix. This means that there exists a decomposition of each unfolding matrix of $f$ of the form

$$f(x_{1:k}, x_{k+1:n}) = \Phi_k(x_{1:k}, \alpha_k)\Psi_k(\alpha_k, x_{k+1:n}), \tag{6}$$

where $\alpha_k \in [r_k]$, and $\Phi_k$ and $\Psi_k$ are matrices whose column and row spaces, respectively, span the column and row spaces of $f$. Since we assume that $f$ admits a TT representation, one possible decomposition of its $k$-th unfolding matrix is obtained by contracting the first $k$ and the last $n-k$ cores, respectively:

$$\begin{aligned}
\Phi_k(x_{1:k}, \alpha_k) &:= G_1(x_1, \alpha_1)G_2(\alpha_1, x_2, \alpha_2)\cdots G_k(\alpha_{k-1}, x_k, \alpha_k), \\
\Psi_k(\alpha_k, x_{k+1:n}) &:= G_{k+1}(\alpha_k, x_{k+1}, \alpha_{k+1})\cdots G_{n-1}(\alpha_{n-2}, x_{n-1}, \alpha_{n-1})G_n(\alpha_{n-1}, x_n),
\end{aligned} \tag{7}$$

for all $k \in [n-1]$. Therefore, if we can efficiently obtain a set of vectors spanning the column space of the unfolding matrices of $f$, we can define the following series of equations:

$$\Phi_k(x_{1:k}, \alpha_k) = \Phi_{k-1}(x_{1:k-1}, \alpha_{k-1})G_k(\alpha_{k-1}, x_k, \alpha_k), \tag{8}$$

from which each core can be determined. This result is formalized in the following proposition, as proved in Ref. [63].

**Proposition 2.1.** *Let $f : [d_1] \times \cdots \times [d_n] \to \mathbb{R}$ a discrete function that admits a TT representation with ranks $r_1, \dots, r_{n-1}$, and define tensors $\Phi_k : [d_1] \times \cdots [d_k] \times [r_k] \to \mathbb{R}$ such that the column space of $\Phi_k(x_{1:k}, \alpha_k)$ is the same as the column space of the $k$-th unfolding matrix of $f$. Then, for the unknowns $G_1 : [d_1] \times [r_1] \to \mathbb{R}$, $G_k : [r_{k-1}] \times [d_k] \times [r_k] \to \mathbb{R}$ for all $k \in \{2, \dots, n-1\}$, and $G_n : [r_{n-1}] \times [d_n] \to \mathbb{R}$, we consider the following series of equations:*

$$\begin{aligned}
G_1(x_1, \alpha_1) &= \Phi_1(x_1, \alpha_1), \\
\Phi_{k-1}(x_{1:k-1}, \alpha_{k-1})G_k(\alpha_{k-1}, x_k, \alpha_k) &= \Phi_k(x_{1:k-1}, x_k, \alpha_k), \quad k \in \{2, \dots, n-1\}, \\
\Phi_{n-1}(x_{1:n-1}, \alpha_{n-1})G_n(\alpha_{n-1}, x_n) &= f(x_{1:n-1}, x_n),
\end{aligned} \tag{9}$$

*which we refer to as the Core Determining Equations (CDEs). Each of these equations has a unique solution, and the solutions $G_1, \dots, G_n$ satisfy*

$$f(x_1, \dots, x_n) = G_1(x_1, \alpha_1)G_2(\alpha_1, x_2, \alpha_2)\cdots G_{n-1}(\alpha_{n-2}, x_{n-1}, \alpha_{n-1})G_n(\alpha_{n-1}, x_n). \tag{10}$$

*Remark* 2.1. When defining the CDEs (9), one possible approach would be to use the first $k-1$ cores obtained on the left-hand side to solve for the $k$-th core. However, introducing the matrices $\Phi_{k-1}$ and $\Phi_k$ on both sides of the equation helps mitigate the accumulation of errors that could arise from reusing the previously computed solutions $G_1, \dots, G_{k-1}$.

To implement this idea efficiently, we need to define two types of sketch functions, that serve distinct purposes. First, similarly to randomized SVD [80], we define random projections $T_{k+1} : [d_{k+1}] \times \cdots \times [d_n] \times [N_{k+1:n}] \to \mathbb{R}$ for $k \in [n-1]$, referred to as *right sketches*. These projections will be used to generate random vectors in the range of the $k$-th unfolding matrix of $f$. Secondly, it should be noted that the CDEs (9) are matrix equations with a number of coefficients that grows exponentially with the dimension $n$, making the system of equations overdetermined and computationally infeasible to solve. To reduce the number of equations,

we define, for $k \in \{2, \ldots, n\}$, the *left sketches* $S_{k-1} : [N_{1:k-1}] \times [d_1] \times \cdots \times [d_{k-1}] \to \mathbb{R}$. Applying these sketches to both sides of Eq. (9), we obtain

$$S_{k-1}(\beta_{k-1}, x_{1:k-1})\Phi_{k-1}(x_{1:k-1}, \alpha_{k-1})G_k(\alpha_{k-1}, x_k, \alpha_k) = S_{k-1}(\beta_{k-1}, x_{1:k-1})\Phi_k(x_{1:k}, \alpha_k), \quad (11)$$

with $k \in \{2, \ldots, n-1\}$.

As will be detailed later in this section, the left sketch functions must have a recursive definition to incorporate the SVD into the algorithm for determining the actual rank $r_k$. Specifically, the sketch $S_k$ is constructed as a composition of the previous sketch, $S_{k-1}$, and an auxiliary function $s_k : [N_{1:k}] \times [N_{1:k-1}] \times [d_k] \to \mathbb{R}$:

$$S_k(\beta_k, x_{1:k}) := S_{k-1}(\beta_{k-1}, x_{1:k-1})s_k(\beta_k, \beta_{k-1}, x_k), \quad (12)$$

for $k \in \{2, \ldots, n-1\}$, and $S_1 := s_1$ with $s_1 : [N_1] \times [d_1] \to \mathbb{R}$.

*Remark* 2.2. Since $N_{a:b}$ denotes the number of unique elements in $\mathbf{x}_{a:b}$, it is clear that it is upper bounded by the total amount of possible values in this set. Specifically, $N_{a:b} \leq d_a \cdots d_b$.

In contrast to the TT-RS approach [63], which employs random sketches, we propose the use of orthogonal projection sketches. Specifically, we define sketches $S_{k-1}$ and $T_{k+1}$ such that applying them to $f$ corresponds to evaluating the function on the subsets of variables $\mathbf{x}_{1:k-1}$ and $\mathbf{x}_{k+1:n}$, respectively, thereby constructing the tensor $f(\mathbf{x}_{1:k-1}, [d_k], \mathbf{x}_{k+1:n})$. This method is efficient for any function, requiring only a polynomial number of function calls, unlike random sketches, whose cost scales exponentially with $n$.

Although our method does not explore the full domain—as do random sketches in TT-RS or `maxvol` in TT-CI—we show that a small set of pivots from a relevant subregion suffices to achieve approximations at least as accurate as those from TT-CI, but at significantly lower cost (see Section 3). For our target applications, vanilla TT-RS is infeasible due to the exponential cost of random sketches. To select suitable pivots, we show that for regular functions, uniformly random points from the domain suffice, while for machine learning models—which are typically sparse tensors, or functions with flat landscapes and narrow peaks, often exhibiting discrete symmetries—using pivots from the training set is more effective. Prior work has also shown that in such settings, `maxvol` may be limited by ergodicity issues, and that incorporating domain knowledge is crucial to guide exploration toward relevant regions [77]. Our approach thus combines the strengths of TT-RS and TT-CI, overcoming the efficiency and exploration limitations of both.

---

**Algorithm 1** Tensor Train via Recursive Sketching from Samples

---

**Require:** Discrete function $f : [d_1] \times \cdots \times [d_n] \to \mathbb{R}$.
**Require:** Sketch samples $\mathbf{x} = \{(\mathbf{x}_1^i, \ldots, \mathbf{x}_n^i) : \mathbf{x}_j^i \in [d_j]\}_{i=1}^N$.
**Require:** Target ranks $r_1, \ldots, r_{n-1}$.

$\quad s_1, \ldots, s_{n-1}, T_2, \ldots, T_n \leftarrow \text{SKETCHBUILDING}(\mathbf{x})$.
$\quad \widetilde{\Phi}_1, \ldots, \widetilde{\Phi}_d \leftarrow \text{SKETCHING}(f, s_1, \ldots, s_{n-1}, T_2, \ldots, T_n)$.
$\quad B_1, \ldots, B_n \leftarrow \text{TRIMMING}(\widetilde{\Phi}_1, \ldots, \widetilde{\Phi}_n, r_1, \ldots, r_{n-1})$.
$\quad A_1, \ldots, A_{n-1} \leftarrow \text{SYSTEMFORMING}(B_1, \ldots, B_{n-1}, s_1, \ldots, s_{n-1})$.
$\quad$ Solve via least-squares for the unknowns $G_1 : [d_1] \times [r_1] \to \mathbb{R}$, $G_k : [r_{k-1}] \times [d_k] \times [r_k] \to \mathbb{R}$ for all $k \in \{2, \ldots, n-1\}$, and $G_n : [r_{n-1}] \times [d_n] \to \mathbb{R}$:

$$G_1(x_1, \alpha_1) = B_1(x_1, \alpha_1),$$
$$A_{k-1}(\beta_{k-1}, \alpha_{k-1})G_k(\alpha_{k-1}, x_k, \alpha_k) = B_k(\beta_{k-1}, x_k, \alpha_k), \quad k \in \{2, \ldots, n-1\},$$
$$A_{n-1}(\beta_{n-1}, \alpha_{n-1})G_n(\alpha_{n-1}, x_n) = B_n(\beta_{n-1}, x_n).$$

$\quad$ **return** $G_1, \ldots, G_n$

---

The complete TT-RSS decomposition is outlined in Algorithm 1, where it is divided into a series of subroutines, following a structure similar to that in Ref. [63]. These subroutines are described in detail in the following section.

## 2.2 Details of subroutines

For clarity in the description of the entire algorithm, we divide it into the following subroutines: SKETCHBUILDING (Section 2.2.1), SKETCHING (Section 2.2.2), TRIMMING (Section 2.2.3), SYSTEMFORMING (Section 2.2.4), and SOLVING (Section 2.2.5).

### 2.2.1 SKETCHBUILDING

The first step of the algorithm involves applying the sketch functions $T_{k+1}$ and $S_{k-1}$ to $f$, forming the coefficient matrices $\widetilde{\Phi}_k$. To ensure efficiency in the sketching process, we define preliminary sketches $P_{a:b} : [N_{a:b}] \times [d_a] \times \cdots \times [d_b] \to \mathbb{R}$, which will be used to construct the left and right sketches, as:

$$P_{a:b}(i, x_{a:b}) := \mathbb{I}_{a:b}(\mathbf{x}^i_{a:b}, x_{a:b}), \tag{13}$$

for $i \in [N_{a:b}]$, where $\mathbb{I}_{a:b} \in \mathbb{R}^{d_a \cdots d_b}$ denotes the identity matrix. This is, $P_{a:b}$ represents the orthogonal projection onto the subspace determined by the variables $x_a$ to $x_b$ of the pivots, $\mathbf{x}_{a:b}$. Similarly, we denote $P^\top_{a:b} : [d_a] \times \cdots \times [d_b] \times [N_{a:b}] \to \mathbb{R}$ as the function whose matricization is the transpose of the matricization of $P_{a:b}$, such that:

$$P^\top_{a:b}(x_{a:b}, i) = P_{a:b}(i, x_{a:b}). \tag{14}$$

The use of these projections to construct the sketches constitutes the main difference from the approach in Ref. [63], which relies solely on random sketching matrices.

From these projections, we can define the left and right sketches as follows:

$$
\begin{aligned}
S_k(\beta_k, x_{1:k}) &:= P_{1:k}(\beta_k, x_{1:k}) = \mathbb{I}_{1:k}(\mathbf{x}^{\beta_k}_{1:k}, x_{1:k}), \\
T_k(x_{k:n}, \gamma_{k-1}) &:= P^\top_{k:n}(x_{k:n}, \eta_{k-1}) U_k(\eta_{k-1}, \gamma_{k-1}) = \mathbb{I}_{k:n}(x_{k:n}, \mathbf{x}^{\eta_{k-1}}_{k:n}) U_k(\eta_{k-1}, \gamma_{k-1}),
\end{aligned}
\tag{15}
$$

where $U_k \in O(N_{k:n})$ is a random orthogonal (or, in the case of complex coefficients, unitary) matrix sampled from the Haar measure as described in Ref. [109]. Note that with this definition, the sketches $S_k$ admit a recursive formulation if we define

$$s_k(\beta_k, \beta_{k-1}, x_k) := \begin{cases} 1, & \text{if } \beta_k = \beta_{k-1}, \text{ and } x_k = \mathbf{x}^{\beta_k}_k, \\ 0, & \text{otherwise,} \end{cases} \tag{16}$$

for $k \in \{2, \ldots, n-1\}$.

*Remark* 2.3. Choosing $U_k$ as an orthogonal random matrix aims to construct the sketches $T_k$ as Johnson-Lindenstrauss embeddings. Specifically, instead of applying the whole $U_k$, one can retain only a subset of its columns, yielding a projection to a lower-dimensional space, which helps reduce the computational cost of subsequent steps—such as the SVD in TRIMMING. Furthermore, since the orthogonal projections $P^\top_{a:b}$ are deterministic matrices that restrict the action of $f$ to the subspace defined by $\mathbf{x}_{a:b}$, the isometry $U_k$ introduces randomness while approximately preserving pairwise distances. No additional isometry is introduced in the left sketches $S_k$, as they are solely intended to reduce the number of equations, and introducing an isometry would be irrelevant for solving them.

### 2.2.2 SKETCHING

Applying these sketches entails first constructing the tensors $\bar{\Phi}_k : [N_{1:k-1}] \times [d_k] \times [N_{k+1:n}] \to \mathbb{R}$ as

$$
\begin{aligned}
\bar{\Phi}_k &:= P_{1:k-1}([N_{1:k-1}], x_{1:k-1}) f(x_{1:k-1}, [d_k], x_{k+1:n}) P_{k+1:n}^{\top}(x_{k+1:n}, [N_{k+1:n}]) \\
&= \mathbb{I}_{1:k-1}(\mathbf{x}_{1:k-1}, x_{1:k-1}) f(x_{1:k-1}, [d_k], x_{k+1:n}) \mathbb{I}_{k+1:n}(x_{k+1:n}, \mathbf{x}_{k+1:n}) \\
&= f(\mathbf{x}_{1:k-1}, [d_k], \mathbf{x}_{k+1:n}), \quad k \in \{2, \dots, n-1\},
\end{aligned}
\tag{17}
$$

that is, evaluating the function $f$ on all the points in the set $\mathbf{x}_{1:k-1} \times [d_k] \times \mathbf{x}_{k+1:n}$, which entails a number of function calls that is polynomial in the number of pivots $N$ (see Section 2.4 for further details). The ability to substitute the projection with only a polynomial number of function calls is what makes our approach efficient, in contrast to Ref. [63], whose random sketches scale exponentially with $n$. From $\bar{\Phi}_k$, we form the tensors $\widetilde{\Phi}_k$ by multiplying with $U_{k+1}$ on the right:

$$
\begin{aligned}
\widetilde{\Phi}_k(\beta_{k-1}, x_k, \gamma_k) &:= \bar{\Phi}_k(\beta_{k-1}, x_k, \eta_k) U_{k+1}(\eta_k, \gamma_k) \\
&= S_{k-1}(\beta_{k-1}, x_{1:k-1}) f(x_{1:k-1}, x_k, x_{k+1:n}) T_{k+1}(x_{k+1:n}, \gamma_k).
\end{aligned}
\tag{18}
$$

Finally, let $\bar{\Phi}_1 := f([d_1], \mathbf{x}_{2:n})$ and $\bar{\Phi}_n := f(\mathbf{x}_{1:n-1}, [d_n])$, thereby defining the corresponding tensors $\widetilde{\Phi}_1$ and $\widetilde{\Phi}_n$.

The tensors $\widetilde{\Phi}_k$ serve the same role as the tensors $\Phi_k$ on the right-hand side of Eq. (9), with the distinction that the indices $\beta_{k-1}$ and $\gamma_k$ range over $[N_{1:k-1}]$ and $[N_{k+1:n}]$, respectively, instead of $[r_{k-1}]$ and $[r_k]$. To form the tensors on the left-hand side of Eq. (9), one would typically contract $f$ with $S_{k-1}$ and $T_k$. However, as will be detailed in the subsequent subroutines, this additional computation can be avoided by reusing the previously computed $\widetilde{\Phi}_{k-1}$ and taking advantage of the recursive definition of $S_{k-1}$.

*Remark* 2.4. As commented earlier, in the general description of TT-RS provided in Ref. [63], left and right sketches are described as random projections, and only the construction of efficient sketches for Markov chains is discussed. However, applying random projections in a space of size $\prod_{i=1}^{n} d_i$ is generally infeasible. Our adaptation, TT-RSS, addresses this challenge by introducing the projections $P_{a:b}$, whose practical computation only requires evaluating $f$ on the selected pivots. This approach reduces the complexity of creating tensors $\bar{\Phi}_k$ to $O(N_{1:k-1} d_k N_{k+1:n})$.

### 2.2.3 TRIMMING

To find the proper ranks $r_k$, we perform SVD on the matrices $\widetilde{\Phi}_k((\beta_{k-1}, x_k), \gamma_k)$ to find functions $B_k : [N_{1:k-1}] \times [d_k] \times [r_k] \to \mathbb{R}$ and $C_k : [r_k] \times [N_{k+1:n}] \to \mathbb{R}$ satisfying

$$
\widetilde{\Phi}_k(\beta_{k-1}, x_k, \gamma_k) = B_k(\beta_{k-1}, x_k, \alpha_k) C_k(\alpha_k, \gamma_k),
\tag{19}
$$

for all $k \in \{2, \dots, n-1\}$. For $k = 1$ we obtain $\widetilde{\Phi}_1(x_1, \gamma_1) = B_1(x_1, \alpha_1) C_1(\alpha_1, \gamma_1)$ by performing SVD on $\widetilde{\Phi}_1(x_1, \gamma_1)$, and finally, we set $B_n(\beta_{n-1}, x_n) := \widetilde{\Phi}_n(\beta_{n-1}, x_n)$.

*Remark* 2.5. The SVD is required only if $N_{k+1:n} > r_k$; otherwise, we set $B_k := \widetilde{\Phi}_k$. However, it is standard and recommended to choose the number of pivots $N$ such that $N_{k+1:n} > r_k$ for all $k \in [n-1]$, in order to effectively capture the range of all the unfolding matrices of $f$. This is particularly important when the selected pivots are not guaranteed to provide the full range of the matrix, as they serve merely as a heuristic.

#### 2.2.4 SYSTEMFORMING

To form the coefficient matrices on the left-hand side of the CDEs (9), we construct the following matrices $A_k$ from $B_k$ and $s_k$:

$$A_k(\beta_k, \alpha_k) := s_k(\beta_k, \beta_{k-1}, x_k) B_k(\beta_{k-1}, x_k, \alpha_k), \quad k \in \{2, \ldots, n-1\}. \tag{20}$$

This highlights the necessity of determining the sketches $S_k$ recursively, as it enables the construction of $A_k$ from the tensor $B_k$, which already has the appropriate rank $r_k$. While one could argue that recursiveness might be avoided if an explicit formulation of $B_k$ from $\widetilde{\Phi}_k$ were provided, such an approach would require prior knowledge of the rank or first performing SVD to subsequently identify an explicit projection that yields the left singular vectors. In the proposed algorithm, left sketches are inherently described recursively by definition, and thus the non-recursive case is disregarded.

#### 2.2.5 SOLVING

Finally, the CDEs (9) can be reformulated in terms of the tensors $A_{k-1}$ and $B_k$ as follows:

$$\begin{aligned} G_1(x_1, \alpha_1) &= B_1(x_1, \alpha_1), \\ A_{k-1}(\beta_{k-1}, \alpha_{k-1}) G_k(\alpha_{k-1}, x_k, \alpha_k) &= B_k(\beta_{k-1}, x_k, \alpha_k), \quad k \in \{2, \ldots, n-1\}, \\ A_{n-1}(\beta_{n-1}, \alpha_{n-1}) G_n(\alpha_{n-1}, x_n) &= B_n(\beta_{n-1}, x_n). \end{aligned} \tag{21}$$

From this reformulation, the cores $G_1, \ldots, G_n$ can be determined via least-squares.

*Remark* 2.6. In high-dimensional scenarios, it is crucial to have a functional approximation $\hat{p}$ of $p$, rather than merely an empirical distribution $\hat{p}_{\mathrm{E}}$. If only $N$ samples, polynomial in $n$, are available, typically there will be no observations in the off-diagonal regions of the sets $\mathbf{x}_{1:k} \times \mathbf{x}_{k+1:n}$, especially for values of $k$ sufficiently far from 1 and $n$. Therefore, when sketching $\hat{p}_{\mathrm{E}}$ as described in this section, the resulting matrices $\widetilde{\Phi}_k$ will essentially be submatrices of the identity multiplied by random matrices $U_{k+1}$. This results in a series of CDEs (9) where both the coefficient and bias matrices, $A_{k-1}$ and $B_k$, are randomly generated, yielding erroneous results. In contrast, with a polynomial number of samples, we could first train a simple machine learning model and use it as the approximation $\hat{p}$ for the decomposition. This approach would enable us to fill in the off-diagonal elements of the matrices $\widetilde{\Phi}_k$, leading to significantly better results.

### 2.3 Continuous case

We adapt the described algorithm to decompose continuous functions, the most common application scenario. Following Ref. [63], we generalize their approach by relaxing the requirement of an orthonormal function basis to represent $f$. Instead, we allow any embedding functions as a basis and aim to identify discrete cores that approximate the function effectively within this framework.

When $f$ is a continuous function, the main argument remains nearly the same, with just a few necessary adaptations. First, let $f : X_1 \times \cdots \times X_n \to \mathbb{R}$, where $X_1, \ldots, X_n \subset \mathbb{R}$. Typically, $X_k = [u, v]$, with $u, v \in \mathbb{R}$, for all $k \in [n]$. It is assumed that $f$ can be approximated by an expansion in a basis of functions formed by the tensor product of one-dimensional functions $\phi_k(x) = (\phi_k^{i_k}(x))_{i_k \in [d_k]}$, referred to as the *embedding functions*. Thus, $f$ is represented as

$$f(x_1, \ldots, x_n) \approx \breve{f}(i_1, \ldots, i_n) \phi_1(i_1, x_1) \cdots \phi_n(i_n, x_n), \tag{22}$$

where $\phi_k(i_k, x_k) = \phi_k^{i_k}(x_k)$, and $\breve{f} : [d_1] \times \cdots \times [d_n] \to \mathbb{R}$ is the tensor of coefficients. The goal is to obtain cores $\breve{G}_1, \ldots, \breve{G}_n$ that provide the TT representation of $\breve{f}$. In this context, the input dimensions $d_k$ of each core $\breve{G}_k$ are referred to as the *embedding dimensions*.

To achieve this, the tensors $\widetilde{\Phi}_k$ are first constructed from evaluations of $f$, similarly to Eq. (17), and the following equations are imposed:

$$\widetilde{\Phi}_k(\beta_{k-1}, x_k, \gamma_k) = \breve{\Phi}_k(\beta_{k-1}, i_k, \gamma_k)\phi_k(i_k, x_k), \tag{23}$$

from which the tensors $\breve{\Phi}_k$ can be obtained. Using these, one can construct the tensors $B_k$ and $A_k$, and solve the CDEs (9).

To determine the functions $\bar{\Phi}_k$, one must define continuous versions of the left and right sketches. This requires defining projections $P_{a:b}$ as the Dirac delta function $\delta$, which satisfies

$$\int_{X_a \times \cdots \times X_b} f(x_{a:b})\delta(x_{a:b} - \mathbf{x}^i_{a:b})dx_{a:b} = f(\mathbf{x}^i_{a:b}). \tag{24}$$

The projections $P_{a:b}$ are thus given by

$$P_{a:b}(i, x_{a:b}) := \delta(x_{a:b} - \mathbf{x}^i_{a:b}), \tag{25}$$

and their application to $f$ yields

$$\bar{\Phi}_k([N_{1:k-1}], x_k, [N_{k+1:n}]) := f(\mathbf{x}_{1:k-1}, x_k, \mathbf{x}_{k+1:n}), \quad k \in \{2, \ldots, n-1\}. \tag{26}$$

As in the finite case, functions $\widetilde{\Phi}_k$ are then constructed by multiplying $\bar{\Phi}_k$ on the right by $U_{k+1}$.

To solve Eq. (23), if the embedding functions $\{\phi_k^{i_k}\}_{i_k \in [d_k]}$ form an orthonormal basis, tensors $\breve{\Phi}_k$ can be computed as

$$\breve{\Phi}_k(\beta_{k-1}, j_k, \gamma_k) = \int_{X_k} \widetilde{\Phi}_k(\beta_{k-1}, x_k, \gamma_k)\phi_k^{j_k}(x_k)dx_k, \tag{27}$$

for all $j_k \in [d_k]$. However, in practice, common embeddings do not form an orthonormal basis, in which case Eq. (23) is solved via least-squares. This involves evaluating $\widetilde{\Phi}_k$ and $\{\phi_k^{i_k}\}_{i_k \in [d_k]}$ at a set of points $\mathbf{y}_k$ to impose the matrix equation

$$\widetilde{\Phi}_k(\beta_{k-1}, \mathbf{y}_k, \gamma_k) = \breve{\Phi}_k(\beta_{k-1}, i_k, \gamma_k)\phi_k(i_k, \mathbf{y}_k). \tag{28}$$

For well-posedness, $\mathbf{y}_k$ must contain at least $d_k$ elements. If the focus is primarily on the subregion defined by the pivots, $\mathbf{y}_k$ can be chosen as $\mathbf{x}_k$. More generally, a set of equidistant points within the domain, such as $\mathbf{y}_k = \{u, u+h, u+2h, \ldots, v\}$, is used, where it is assumed that $X_k = [u, v]$ and $h$ is a suitable step size to ensure a good approximate solution while maintaining computational efficiency. We will refer to this additional subroutine as FITTING, which is executed after SKETCHING.

Performing SVD on $\breve{\Phi}_k$ yields tensors $\breve{B}_k$, and tensors $\breve{A}_k$ are constructed as in Eq. (20). However, a subtlety must be addressed when defining the auxiliary functions $s_k$. To provide a recursive definition of the left sketches, one can proceed similarly to the discrete case and define functions $s_k : [N_{1:k}] \times [N_{1:k-1}] \times X_k \to \mathbb{R}$ as

$$s_k(\beta_k, \beta_{k-1}, x_k) := \begin{cases} \delta(x_k - \mathbf{x}^{\beta_k}_k), & \text{if } \beta_k = \beta_{k-1}, \\ 0, & \text{otherwise.} \end{cases} \tag{29}$$

This definition aligns with the discrete formulation of $B_k$ used to construct $A_k$. However, since Eq. (23) effectively "removes the embedding", to appropriately compute the evaluations of $f$ the embedding must be reintroduced. Thus, we need auxiliary functions $\check{s}_k : [N_{1:k}] \times [N_{1:k-1}] \times [d_k] \to \mathbb{R}$, defined as

$$\check{s}_k(\beta_k, \beta_{k-1}, i_k) := \begin{cases} \phi_k(i_k, x_k)\delta(x_k - \mathbf{x}^{\beta_k}_k), & \text{if } \beta_k = \beta_{k-1}, \\ 0, & \text{otherwise.} \end{cases} \tag{30}$$

Finally, $\breve{A}_k$ is constructed as

$$\breve{A}_k(\beta_k, \alpha_k) := \breve{s}_k(\beta_k, \beta_{k-1}, i_k)\breve{B}_k(\beta_{k-1}, i_k, \alpha_k). \tag{31}$$

The tensors for the boundary cases $k = 1$ and $k = n$ are constructed analogously, and the CDEs (9) are solved to obtain cores $\breve{G}_1, \ldots, \breve{G}_n$.

The continuous version of TT-RSS is implemented in the TensorKrowch package [66].

## 2.4 Complexity

Let $d = \max_{1 \le k \le n} d_k$. Assuming that the complexity of evaluating $f$ is constant and the cardinality of $\mathbf{y}_k$ is a multiple of $d$ for all $k \in [n]$, the SKETCHING step requires $O(N^2 dn)$ evaluations. Here, as per the notation used earlier, $N = \max_{1 \le a \le b \le n} N_{a:b}$. This complexity includes the SKETCHBUILDING step, since building sketches is implicitly achieved through the evaluations of $f$ on the selected partitions of pivots—namely, on sets of the form $\mathbf{x}_{1:k-1} \times \mathbf{y}_k \times \mathbf{x}_{k+1:n}$, which contain $N_{1:k-1}d_k N_{k:n}$ elements, for all $k \in [n]$. Solving the least-squares problems in FITTING (23), by multiplying by the pseudoinverse of the coefficient matrices, can be completed in $O((d + N^2)d^2 n)$ time. The SVD in TRIMMING is performed in $O(N^3 dn)$ time. Denoting $r = \max_{1 \le k \le n-1} r_k$, the matrix multiplications in SYSTEMFORMING to construct matrices $A_k$ involve $O(N^2 drn)$ operations. Finally, SOLVING the CDEs (9) requires $O(Nr^2(d + 1)n)$ time. Altogether, the overall complexity is

$$O\left(N^2 dn + (d + N^2)d^2 n + N^3 dn + N^2 drn + Nr^2(d + 1)n\right), \tag{32}$$

which is linear in the number of variables $n$. If $N \gg d$, as is typically the case, the dominant term will be $N^3 dn$, making the complexity cubic in $N$.

## 2.5 Relation with TT-CI

With the choice of sketches described in this section, the close relation of our approach with TT-CI [69] becomes evident. As stated at the beginning of the section, we assume that the unfolding matrices of $f$ admit a decomposition as in Eq. (6). Using the skeleton decomposition [71], this can be expressed as

$$f(x_{1:k}, x_{k+1:n}) = f(x_{1:k}, \mathbf{x}_{k+1:n})f(\mathbf{x}_{1:k}, \mathbf{x}_{k+1:n})^{-1}f(\mathbf{x}_{1:k}, x_{k+1:n}). \tag{33}$$

*Remark* 2.7. In our setting, $\mathbf{x}_{a:b}$ denotes the subset formed by the variables $x_a$ to $x_b$ of the fixed set of pivots $\mathbf{x}$, and thus the number of unique elements $N_{a:b}$ could vary for each interval. In TT-CI, these partial pivots $\mathbf{x}_{a:b}$, also referred to as *interpolation sets*, are determined independently by a combinatorial optimization procedure, allowing us to fix $N_{a:b} = N$ for all intervals $a:b$.

As in our approach, the cross interpolation technique can be applied iteratively on the left and right factors of the right-hand side of Eq. (33) to obtain a TT representation. Consequently, the cores can be computed as follows:

$$\begin{aligned}
G_1 &= f([d_1], \mathbf{x}_{2:n}), \\
G_k &= f(\mathbf{x}_{1:k-1}, \mathbf{x}_{k:n})^{-1}f(\mathbf{x}_{1:k-1}, [d_k], \mathbf{x}_{k+1:n}), \quad k \in \{2, \ldots, n-1\}, \\
G_n &= f(\mathbf{x}_{1:n-1}, \mathbf{x}_n)^{-1}f(\mathbf{x}_{1:n-1}, [d_n]),
\end{aligned} \tag{34}$$

which closely resembles Eq. (21), with the only difference being that these equations are solved directly by taking inverses. This underscores how our approach builds upon the ideas inherent in TT-CI, and how this connection can guide the design of better heuristics for selecting pivots.

Although we do not formally establish properties that the selected pivots must satisfy to yield good results (this is, however, an important question that we leave for future work), our proposal of choosing as pivots points from the training set, combined with the defined left and right sketches, aligns with key results from Refs. [69,73,75,77].

The maximum-volume principle [74] states that interpolation sets yielding the best approximation are those for which the intersection matrix has maximum volume. In our study cases, these interpolation sets are formed from points with high probability. Consequently, the matrices $p(\mathbf{x}_{1:k}, \mathbf{x}_{k+1:n})$ tend to have large diagonal values and smaller off-diagonal values. For instance, consider a distribution over a set of images: points $(\mathbf{x}_{1:k}^i, \mathbf{x}_{k+1:n}^j)$, with $i \neq j$, are similar to $\mathbf{x}^i$ when $k$ is close to $n$, and more similar to $\mathbf{x}^j$ when $k$ is close to 1. These off-diagonal points will still have relatively high probability. However, most images will be noisy mixtures of $\mathbf{x}^i$ and $\mathbf{x}^j$, and thus will have low probabilities. Additionally, using random matrices $U_k$ with orthonormal rows helps preserve this high volume.

Moreover, Ref. [73] demonstrated that the TT approximation of $f$ obtained from Eq. (34) is an exact interpolation of the pivots $\mathbf{x}$ if the interpolation sets $\mathbf{x}_{a:b}$ are nested, i.e., if they satisfy $\mathbf{x}_{1:k} \subset \mathbf{x}_{1:k-1} \times [d_k]$ and $\mathbf{x}_{k:n} \subset [d_k] \times \mathbf{x}_{k+1:n}$. This condition is inherently satisfied in our approach, as the interpolation sets are constructed from fixed pivots. In contrast, TT-RSS does not yield an exact interpolation, since the approximation is ultimately determined by the SVD in the TRIMMING subroutine rather than by a cross interpolation decomposition. This is beneficial for our purposes, as we aim to accurately approximate the density $p$, and producing exact values of the model $\hat{p}$ could degrade both the accuracy and privacy of the TT-RSS approximation due to overfitting issues.

Therefore, the TT-RSS algorithm proposed in this paper, while closely resembling TT-CI, differs fundamentally in its purpose: instead of seeking an accurate approximation of a model $\hat{p}$, TT-RSS aims to construct a TT representation of the underlying distribution $p$. This distinction is essential, as tensorizing intricate models may result in a TT model that, while differing significantly from the original model $\hat{p}$ due to potential mismatches in expressive capacity, still provides a good approximation of $p$. TT-RSS achieves this efficiently through a single sweep, performing SVDs to reveal ranks and employing random projections $U_k$ to reduce variance. In contrast, TT-CI aims to accurately approximate $\hat{p}$ through iterative optimization with multiple sweeps to refine interpolation sets, with its rank-revealing adaptation further increasing computational cost via two-site sweeps [69].

Furthermore, as previously discussed in Section 2.1, the `maxvol` step in TT-CI may suffer from ergodicity issues [77], failing to identify good representatives of the function's support in cases of high sparsity, narrow peaks, or discrete symmetries—situations commonly encountered in high-dimensional machine learning tasks. This limitation renders the TT-CI approach insufficient for the problems we aim to address. On the other hand, even when results from TT-RSS are unsatisfactory, rather than iterating with `maxvol` to refine interpolation sets, we can optimize all cores using gradient descent methods, starting from the TT-RSS result, to find better approximations of $p$. This approach motivates a general initialization method for TTs that aligns with the training set distribution and selected embedding, as we detail in Section 5.1.

## 3 Performance

The proposed algorithm, TT-RSS, can be applied to decompose probability distributions $p$, as well as more general functions, since the methods described do not rely on the non-negativity or normalization of $p$ at any step. In this section, we present a series of experiments where TT-RSS is applied to various types of functions. The results demonstrate that our algorithm achieves promising outcomes across all tested scenarios.

Before detailing each example, we must define some elements that are used in the practical implementation of the algorithm. First, to determine the ranks via SVD, we introduce two parameters that bound the number of singular vectors retained. On one hand, we set an upper bound $r$ for all the ranks $r_k$, where $k \in [n-1]$. To account for situations where the actual rank $r_k$ is smaller, we define a parameter $c \in [0,1]$, referred to as the *cumulative percentage*. Using this parameter, we define $\tilde{r}_k$ as the smallest integer satisfying:

$$\frac{\sum_{i=1}^{\tilde{r}_k} \sigma_i}{\sum_j \sigma_j} \geq c \,, \tag{35}$$

where $\sigma_i$ are the singular values. The ranks are then determined as $r_k = \min(\tilde{r}_k, r)$.

To evaluate the accuracy of the results, we compute the relative error of the outputs obtained from the TT approximation provided by TT-RSS, compared to the exact outputs of the functions. Specifically, if $\mathbf{s}$ is a set of $M$ points and we consider $f(\mathbf{s})$ as a vector in $\mathbb{R}^M$, the relative error is computed as:

$$R(\mathbf{s}) := \frac{\left\| f(\mathbf{s}) - \hat{f}_{\text{TT}}(\mathbf{s}) \right\|_2}{\|f(\mathbf{s})\|_2 + \varepsilon} \,, \tag{36}$$

where $\varepsilon$ is a small value added to avoid division by zero. The set $\mathbf{s}$ will usually be referred to as *test samples*, even when the functions to be tensorized are not densities. Relative errors will also be evaluated at the pivots, enabling a comparison between the errors at the points used for tensorizing the functions and those at previously unseen points.

All experiments[1] presented in this work were conducted on an Intel Xeon CPU E5-2620 v4 with 256GB of RAM and an NVIDIA GeForce RTX 3090, using the implementation of TT-RSS available in the TensorKrowch Python package [66].

## 3.1 Non-densities

We begin by tensorizing functions that are not densities. In the two study cases presented here, we consider discrete functions where all input dimensions are equal, i.e., $d_k = d$ for all $k \in [n]$. Consequently, these functions take the form $f : [d]^n \to \mathbb{R}$. However, the algorithm employed in all scenarios is the continuous version of TT-RSS. Notably, this continuous version naturally reduces to the discrete case by adopting the embedding functions $\phi^j(i) = \delta_{ij}$ for $i, j \in [d]$, with $\delta_{ij}$ denoting the Kronecker delta. Here, the subscripts $k$ are omitted because the same embedding is applied to all input variables. Thus, $\phi$ maps the elements of $[d]$ to the canonical basis vectors of $\mathbb{R}^d$, $\{e_1, \ldots, e_d\}$. We refer to this embedding as the *trivial embedding*.

With this setup, the matrix $\phi(i, j)$ appearing on the right-hand side of Eq. (23) becomes the identity matrix, and the equation is automatically satisfied.

### 3.1.1 Random TT

The first scenario we consider involves a function with an exact TT representation. Specifically, we define functions $f_{\text{TT}}^n : [d]^n \to \mathbb{R}$ as TTs, where $n \in \{100, 200, 500\}$ represents, as before, the length of the chain, or the number of variables. In all cases, we fix $d = 2$ and set a constant rank $\bar{r} = 10$. To randomly initialize the cores while avoiding numerical instability (e.g., exploding or vanishing outputs), we construct tensors $G_k$ as stacks of orthogonal matrices. For each $x_k \in [2]$, the matrix $[G_k(\alpha_{k-1}, x_k, \alpha_k)]_{\alpha_{k-1}, \alpha_k \in [\bar{r}]}$ is orthogonal. Consequently, $[G_1(x_1, \alpha_1)]_{\alpha_1 \in [\bar{r}]}$ and $[G_n(\alpha_{n-1}, x_n)]_{\alpha_{n-1} \in [\bar{r}]}$ are unit-norm vectors. As a result, $f_{\text{TT}}^n(x_1, \ldots, x_n)$ represents the scalar product of two unit-norm vectors, ensuring that all outputs lie in the range $[-1, 1]$.

---

[1]The code for the experiments is publicly available at: https://github.com/joserapa98/tensorization-nns.



Using TT-RSS, we reconstruct an approximation $\hat{f}_{\mathrm{TT}}^n$ to $f_{\mathrm{TT}}^n$, varying the number of pivots $N$ from 10 to 35. The pivots $\mathbf{x}$ for the decomposition are selected as random configurations sampled uniformly from the domain $[2]^n$. Additionally, we use a different set $\mathbf{s}$ of $M = 1\,000$ test samples to evaluate the reconstruction error. For all experiments, we set $r = N$, as it is assumed that the true rank $\bar{r}$ is unknown and therefore increasing the allowed rank would yield better approximations. Furthermore, we use a very small value for the cumulative percentage parameter, $c = 1 - 10^{-5}$. Since the algorithm's performance can be influenced by multiple sources of randomness, we repeat the decomposition 10 times for each configuration. The results are shown in the top row of Fig. 1.

In this case, since $f_{\mathrm{TT}}^n$ already has a TT representation with cores $G_k$, the goal is to find a new representation, $\hat{f}_{\mathrm{TT}}^n$, with cores $\widehat{G}_k$, that represent the same TT. As shown in the top-center plot in Fig. 1, the relative error approaches zero as the number of pivots increases. Note that the errors displayed in the figure represent the mean values of the 10 decompositions performed. However, in many instances, the errors $R(\mathbf{s})$ are less than $10^{-14}$. In such cases, we verify that the fidelity between $f_{\mathrm{TT}}^n$ and $\hat{f}_{\mathrm{TT}}^n$, defined as the scalar product of the normalized tensors,

$$F(f, g) = \frac{f(x_1, \ldots, x_n) g(x_1, \ldots, x_n)}{\|f\|_2 \|g\|_2}, \tag{37}$$

equals 1, confirming that the reconstructed TT from TT-RSS is identical to the original. Note that in the above definition, $\|f\|_2$ represents the 2-norm of $f$ viewed as a vector in $\mathbb{R}^{d^n}$.

Finally, we observe that the errors at the pivots are exceptionally low—often several orders of magnitude smaller than the errors at other samples. Only when the original TT is exactly recovered are the errors at the test samples comparable to those at the pivots. This behavior arises because using a small number of pivots for decomposition leads to overfitting at the pivots unless the TT has sufficient expressive power to approximate the entire function accurately. This phenomenon could raise privacy concerns in machine learning models, as overfitting at the pivots might inadvertently reveal sensitive information. This issue can often be mitigated by training the model for a few additional epochs on a larger dataset, as will be discussed in subsequent sections on neural network examples (see Section 3.2.2 and Section 4.1.3).

### 3.1.2 Slater functions

In this experiment, we aim to approximate quantized continuous functions that may not have an exact TT representation. To allow for comparison with prior TT decompositions, we consider the Slater functions, as in Ref. [110]. These functions are defined as:

$$f(z) = \frac{e^{-\|z\|_2}}{\|z\|_2}, \tag{38}$$

where $z \in [0, L]^m$. To quantize the function, we discretize the domain by partitioning each variable $z_i$ into $2^l$ steps. This allows us to define a discrete version of $f$, denoted $f^l : (\{0, 1\}^l)^m \to \mathbb{R}$, as $f^l(x_{1,1}, \ldots, x_{1,l}, \ldots, x_{m,1}, \ldots, x_{m,l}) := f(z_1, \ldots, z_m)$, where

$$z_i = L \sum_{j=1}^{l} x_{i,j} 2^{-j}. \tag{39}$$

Although the domain here is $\{0, 1\}$ instead of $[d]$, we can redefine the domain by shifting it by one unit. Specifically, we may define $\bar{f}^l : (\{1, 2\}^l)^m \to \mathbb{R}$ as $\bar{f}^l(x) := f^l(x - \mathbf{1})$, where $\mathbf{1}$ is the all-ones vector. For simplicity, we typically omit this distinction in the discussion.

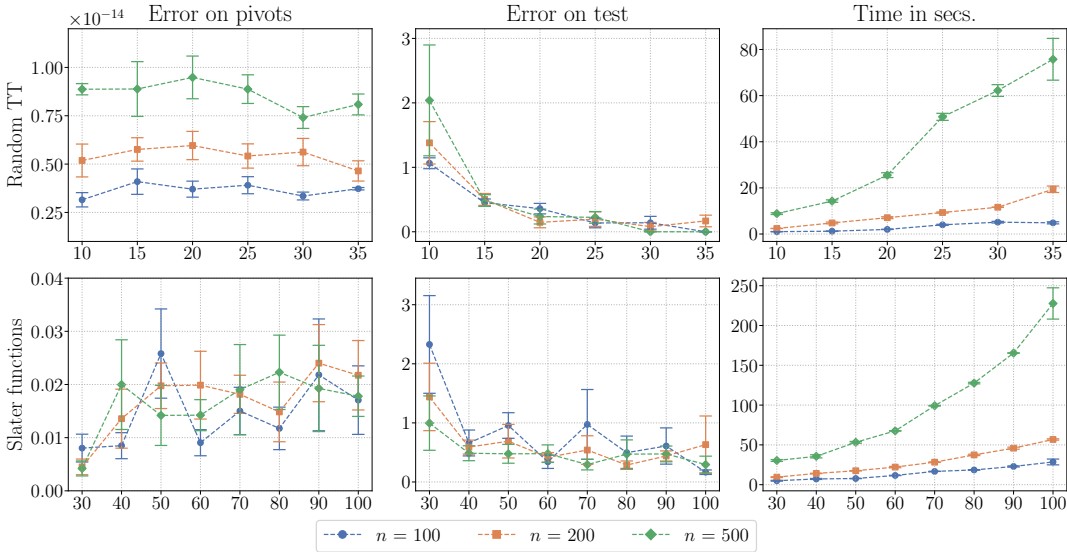

Figure 1: Performance of TT-RSS applied to different functions when varying the number of pivots $N$ (x-axis). From left to right, the columns show: the relative error on the $N$ pivots $\mathbf{x}$ used in TT-RSS, $R(\mathbf{x})$; the relative error on a set of $M$ test samples $\mathbf{s}$ from the functions' domains, $R(\mathbf{s})$; and the time (in seconds) required to perform the decompositions. For each value of $N$, the decomposition is performed 10 times. The figures display the mean values with error bars at $\pm 0.5\sigma$, where $\sigma$ denotes the standard deviation. Additionally, three configurations with different numbers of variables, $n = 100$, $n = 200$, and $n = 500$, are shown in different colors. All computations are performed in double precision.

As in the previous example, we consider an input dimension $d = 2$. To analyze functions of lengths $n \in \{100, 200, 500\}$, since $n = ml$, we fix $m = 5$ and choose $l \in \{20, 40, 100\}$, so that we increase the number of discretization steps without changing the number of variables $m$. In all cases, we set $L = 10$, with a maximum rank of $r = 10$ and $c = 1 - 10^{-5}$. From the uniform distribution over the domain, we select $N \in \{30, 40, \ldots, 100\}$ pivots, along with $M = 1\,000$ samples for testing.

Although these results are less conclusive than those of the previous case, the bottom row of Fig. 1 shows that the errors decrease as $N$ increases. Remarkably, even with a higher discretization level of $l = 100$, we achieve test errors on the order of $10^{-1}$ for $N \geq 80$, with computational times ranging between 125 and 250 seconds. As observed earlier, the errors at the pivots are generally smaller than those at other samples. However, since the Slater function does not have an exact TT representation, this behavior is expected to persist unless higher ranks $r$ are allowed, enabling better approximations across all input configurations.

Additionally, it is noteworthy that errors at the pivots tend to increase slightly as $N$ grows. This trend suggests that as the approximation improves for the entire function, the tendency to overfit the pivots diminishes. This observation aligns with the expectation that larger numbers of pivots and increased expressive power allow for more balanced approximations.

### 3.1.3 Comparison with TT-CI

For comparison with prior tensorization techniques, Table 1 shows the performance of TT-RSS and TT-CI—using the implementation from the `tntorch` package [111]—applied to the previously described functions, under a reduced set of configurations. Specifically, we tensorize random TTs for $n \in \{100, 200\}$ and Slater functions for $l \in \{20, 40, 100\}$ (equiva-

lently, $n \in \{100, 200, 500\}$), using two pivot set sizes: $N \in \{20, 35\}$ for random TTs, and $N \in \{50, 100\}$ for Slater functions. For TT-CI, we perform a single round of sweeps along the TT chain—left-to-right and right-to-left—and, importantly, we specify the TT ranks in advance: $r = \bar{r}$ for random TTs, and $r = 10$ for Slater functions. That is, TT-CI is given an advantage to yield results that are more directly comparable.

As noted in previous subsections, the randomness in TT-RSS, mainly due to pivot selection, can cause significant variability in results. For instance, with $N = 20$ pivots in random TTs, test errors typically concentrate around $10^{-14}$ when the TT is exactly recovered, or around $10^{-1}$ when it is not. Thus, while Fig. 1 is useful to observe general trends in error and time as the number of pivots increases, it does not reflect typical performance. To provide a fairer comparison, Table 1 reports median values: specifically, the median test errors and median tensorization times (in seconds). For TT-CI, which exhibits less variability due to its more exhaustive optimization, the medians closely match the means. The time costs shown for TT-CI correspond to the sweeps, with the full algorithm time—including pivot preparation and decomposition setup—given in parentheses. For TT-RSS, we report the total time for the full

Table 1: Performance comparison of TT-RSS and TT-CI applied to different functions with varying numbers of variables $n$ and pivots $N$. The table reports the relative error on a set of $M$ test samples $\mathbf{s}$ from the functions' domains, $R(\mathbf{s})$, and the time (in seconds) required to perform the decompositions. For TT-CI, the reported time corresponds to the sweeps, while the value in parentheses includes the total time of the full algorithm, including preliminary steps for decomposition setup. Each decomposition is repeated 10 times, and the median values are shown. All computations are performed in double precision.

| $f$ | $n$ | Tensorization algorithm | Error on test | Time in secs. |
|---|---|---|---|---|
| Random TT | 100 | TT-RSS ($N = 20$) | $3.60 \times 10^{-1}$ | $1.97 \times 10^{0}$ |
| | | TT-RSS ($N = 35$) | $4.26 \times 10^{-15}$ | $4.25 \times 10^{0}$ |
| | | TT-CI | $7.56 \times 10^{-15}$ | $3.83 \times 10^{1}$ ($4.73 \times 10^{2}$) |
| | 200 | TT-RSS ($N = 20$) | $9.23 \times 10^{-15}$ | $7.08 \times 10^{0}$ |
| | | TT-RSS ($N = 35$) | $6.56 \times 10^{-15}$ | $1.89 \times 10^{1}$ |
| | | TT-CI | $1.08 \times 10^{-14}$ | $1.37 \times 10^{2}$ ($2.53 \times 10^{3}$) |
| Slater functions | 100 | TT-RSS ($N = 50$) | $8.06 \times 10^{-1}$ | $7.70 \times 10^{0}$ |
| | | TT-RSS ($N = 100$) | $1.58 \times 10^{-1}$ | $2.76 \times 10^{1}$ |
| | | TT-CI | $6.27 \times 10^{-1}$ | $4.09 \times 10^{1}$ ($4.71 \times 10^{2}$) |
| | 200 | TT-RSS ($N = 50$) | $3.69 \times 10^{-1}$ | $1.76 \times 10^{1}$ |
| | | TT-RSS ($N = 100$) | $2.12 \times 10^{-1}$ | $5.57 \times 10^{1}$ |
| | | TT-CI | $5.63 \times 10^{-1}$ | $1.34 \times 10^{2}$ ($2.53 \times 10^{3}$) |
| | 500 | TT-RSS ($N = 50$) | $3.61 \times 10^{-1}$ | $5.22 \times 10^{1}$ |
| | | TT-RSS ($N = 100$) | $2.29 \times 10^{-1}$ | $1.94 \times 10^{2}$ |
| | | TT-CI | $6.79 \times 10^{-1}$ | $3.30 \times 10^{2}$ ($1.32 \times 10^{4}$) |

algorithm, which involves a single left-to-right sweep.

In this setting, TT-RSS achieves lower errors in significantly less time. As already shown in Fig. 1, increasing the number of pivots reduces the approximation error, yet in some cases, even with the smallest pivot sets, TT-RSS yields lower errors than TT-CI, with notably better time performance. This is not due to running only a single sweep in TT-CI: across all experiments in this work, we observed that once the number of pivots is fixed, TT-CI typically achieves good approximations in a single sweep, and additional sweeps do not significantly improve accuracy. We thus conclude that our pivot selection strategy is sufficient to produce results comparable to TT-CI's more exhaustive search, while requiring significantly less time for decomposition—especially when compared to the full runtime of TT-CI (shown in parentheses), which often increases by one or two orders of magnitude, rendering it impractical for higher-dimensional functions.

Additionally, as previously mentioned, TT-CI is given a strong advantage here by assuming prior knowledge of the TT ranks. In practice, ranks are usually unknown and must be determined adaptively, as is done in TT-RSS via the SVD. In contrast, TT-CI typically uses a fixed number of pivots per sweep, optimized via the `maxvol` routine. If the resulting error is too large, the number of pivots—and hence the rank—is increased in subsequent sweeps to enhance expressivity. A realistic use of TT-CI would thus involve multiple sweeps until the desired number of pivots—and hence, the target accuracy—is achieved, substantially increasing the total runtime. Although recent work has proposed rank-revealing versions of TT-CI [77], TT-RSS remains a more favorable alternative, achieving comparable—or in most cases superior—results in a single, efficient left-to-right sweep.

## 3.2 Neural networks

In this section, the functions decomposed are NNs trained on two widely-known toy datasets: Bars and Stripes, and MNIST [64]. In both cases, we train the models as classifiers. This is, they do not model the entire distribution of the datasets $p(x)$, but rather return a conditioned probability $p(y|x)$ of the class $y$, when provided with an input image $x$. Even though this is not by itself a density, since it is not normalized across all the values of $x$, we can still consider it as a probability distribution. In fact, the TT models we consider in machine learning are usually not normalized, since in these high-dimensional settings, the probability assigned to each sample would typically be smaller than machine precision. However, when normalization needs to be considered, partition functions can be efficiently computed for TT models. Therefore, we will commonly consider non-normalized functions, or simply any non-negative function, as a density. In this case, we will write $p(x_1, \ldots, x_n, y)$ to denote the density. To indicate that we are decomposing a NN approximation, we will denote it by $\hat{p}_{\text{NN}}$.

*Remark* 3.1. Since the TT models have a one-dimensional topology, the position of the variable $y$ in the TT chain might affect the resultant models, due to the different distances between $y$ and the input variables, which affect how correlated these can be [112, 113]. In the experiments, we position the variable $y$ in the middle of the chain.

Furthermore, to represent probabilities with the TT representation, the Born rule is commonly used. This is, the probability is given by the square modulus of the amplitude, or simply the value, associated with an input configuration. Thus, instead of directly approximating $\hat{p}_{\text{NN}}$, we approximate $\hat{q}_{\text{NN}} = \sqrt{\hat{p}_{\text{NN}}}$.

For both datasets, we use the same architecture for feedforward fully connected neural networks: two hidden layers of 500 and 200 neurons, respectively, and an output layer with 2 neurons for the Bars and Stripes dataset, and 10 for MNIST, corresponding to the number of

classes. Finally, the output is passed through the softmax function, defined as

$$\text{softmax}(z)_y = \frac{e^{z_y}}{\sum_{\tilde{y}} e^{z_{\tilde{y}}}}, \tag{40}$$

where $z$ denotes the vector of outputs and $y$ is the class label. This transforms the output of the NN into the conditional probability $\hat{p}_{\text{NN}}(y|x)$. Moreover, in both datasets, we reduce the size of the input images to $12 \times 12$, $16 \times 16$, and $20 \times 20$ pixels, resulting in three cases for $n$: 144, 256, and 400.

In both cases, pivots are selected from the corresponding training sets, with $N$ ranging from 30 to 100. The test samples used to measure accuracy are taken from the test sets, fixing $M = 1000$, as in previous experiments. Additionally, all the TTs are set to have a fixed rank $r = 10$, with $c = 1 - 10^{-5}$.

Finally, apart from measuring the relative errors, we evaluate how different the reconstructed TT models are from the original NNs by measuring, over the $M$ test samples, the percentage of classifications given by the TTs that differ from those given by the NNs. This is a simple metric that indicates, in a more faithful manner than the relative error, how similarly both models perform in practice. Although this does not directly measure the accuracy of the TT models, in these particular examples, where the accuracy of the NN models is close to 100% (100% test accuracy on Bars and Stripes, and 98% on MNIST), the accuracy of the TTs can be approximated as $100 - $ (percentage of different classifications). The results are depicted in Fig. 2.

### 3.2.1 Bars and stripes

This dataset consists of square black-and-white images, which can be of two types: vertical bars or horizontal stripes, depending on the orientation of the interleaving black and white lines. The widths of the lines may vary. For each $n$, we create a dataset with images of varying sizes and train the described NNs using the Adam optimizer, achieving 100% accuracy in both training and testing.

Since the input is binary, we consider this NN as a discrete function and apply the TT-RSS algorithm in the same way as in Section 3.1. In this case, $d = 2$, including the dimension of the $y$ output variable, which also has 2 classes.

For this scenario, we observe in the top row of Fig. 2 that errors on pivots and test samples are very similar, which might indicate that the TT models generalize well. This could be due to the fact that the dataset, even though two-dimensional, is very simple, which might mitigate difficulties for a model with a one-dimensional topology. It can also be seen that the reconstructed TT models achieve accuracies between 90% and 100% in most cases. As the size of the images grows, the models experience a larger drop in accuracy. This might also be influenced by the fact of having fixed the same maximum rank $r = 10$ for all models, which may reduce the expressive power of the larger models.

### 3.2.2 MNIST

Even though this dataset can still be considered a toy dataset, it allows us to demonstrate the performance of the obtained TT models in a slightly more realistic scenario. In this case, each pixel is not binary but is considered as a continuous value between 0 and 1. Thus, a non-trivial continuous embedding is required for the input variables. Inspired by the use of TTs as approximations of polynomials, as presented in Ref. [41], we use the embedding $\phi^i(x) = x^{(i-1)}$, for $i \in [d]$, which we refer to as the *polynomial embedding*. Using these functions, the TT models represent polynomials of the variables $x_1, \ldots, x_n$ up to degree $d-1$ in each variable, where the coefficients are given by the contraction of the TT. This is, the vector

of coefficients is represented as a TT. In this experiment, we keep $d = 2$, as in the previous experiments, which allows us to represent polynomials of degree 1 in each variable. However, in this case, the dimension of the output variable $y$ is 10, corresponding to the number of classes in the dataset. Thus, for the position in the middle of the TT where the output variable is placed, we have $d_{\lfloor n/2 \rfloor} = 10$. Consequently, since this variable is discrete, we use the trivial embedding for that position.

The results of this experiment (in the bottom row of Fig. 2), despite experiencing more significant drops in accuracy, showcase the expected behavior for more general and intricate NN models. Specifically, the expressive power of the tensorized model may differ significantly from that of the NN model, and thus we do not expect to achieve low approximation errors on test samples. Clearly, in situations where the TT model is unable to reproduce the same kind of correlations observed from evaluating the NN on the pivots, the TT will provide a different approximation to the underlying distribution. Therefore, while errors evaluated on output values may be high, the tensorized models might still yield reasonably good accuracies.

We observe that the TT models achieve reasonable accuracies between 75% and 80%, taking only from 20 to 60 seconds in many cases. While this is far from what can be achieved through gradient descent optimization, the results from TT-RSS can serve as starting points for the optimization process. To achieve even better accuracies, one could also increase the allowed rank $r$ and use more pivots $N$, which would correspondingly increase the computation time.

As mentioned earlier, the low errors encountered on pivots might lead to overfitting, which could also raise concerns about generalization and privacy. However, two points must be considered: (i) when forming the tensors $p(\mathbf{x}_{1:k-1}, \mathbf{y}_k, \mathbf{x}_{k+1:n})$, $dN^2$ points are used to tensorize $p$ (in contrast with the $N$ pivots), which helps improve generalization, and (ii) having low

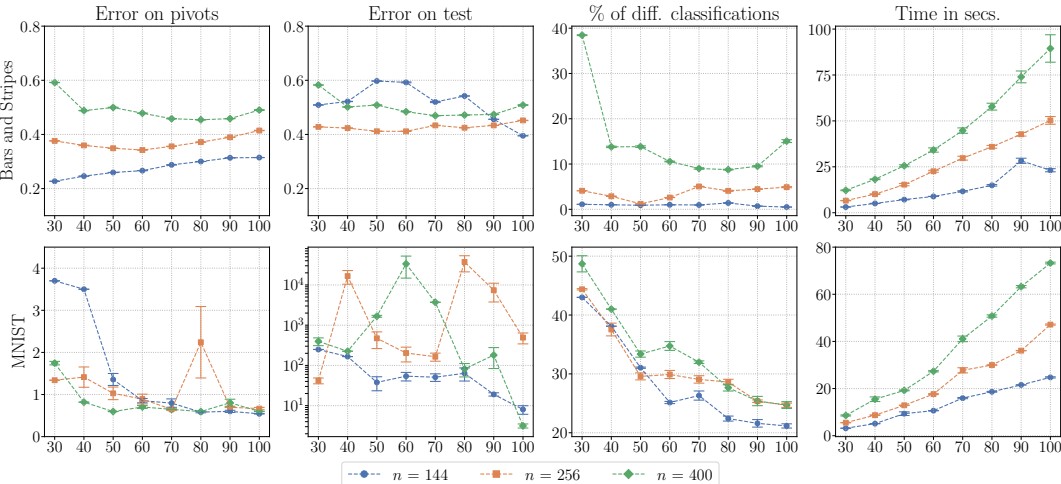

Figure 2: Performance of TT-RSS applied to different NN models when varying the number of pivots $N$ (x-axis). From left to right, columns show: relative error on the $N$ pivots $\mathbf{x}$ used in TT-RSS, $R(\mathbf{x})$; relative error on a set of $M$ test samples $\mathbf{s}$ from the test sets, $R(\mathbf{s})$; percentage of classifications of the TT models that differ from those of the original NN models; and time in seconds taken to perform the decompositions. For each value of $N$, the decomposition is performed 10 times, displaying in the figures the mean values with error bars at $\pm 0.5\sigma$, where $\sigma$ denotes the standard deviation. Also, three configurations are considered with different numbers of variables, $n = 144$, $n = 256$, and $n = 400$, which are displayed using different colors. All computations are performed in double precision.

Table 2: Performance comparison of TT-RSS and TT-CI applied to different NN models with varying numbers of variables $n$ and pivots $N$. The table reports the relative error on a set of $M$ test samples **s** from the test sets, $R(\mathbf{s})$; the percentage of classifications of the TT models that differ from those of the original NN models; and the time (in seconds) required to perform the decompositions. For TT-CI, the reported time corresponds to the sweeps, while the value in parentheses includes the total time of the full algorithm, including preliminary steps for decomposition setup. Each decomposition is repeated 10 times, and the median values are shown. All computations are performed in double precision.

| $f$ | $n$ | Tensorization algorithm | Error on test | % of diff. classifications | Time in secs. |
|---|---|---|---|---|---|
| Bars and Stripes | 144 | TT-RSS ($N = 50$) | $5.97 \times 10^{-1}$ | 0.9% | $7.03 \times 10^{0}$ |
| | | TT-RSS ($N = 100$) | $3.95 \times 10^{-1}$ | 0.5% | $2.29 \times 10^{1}$ |
| | | TT-CI | $4.07 \times 10^{-1}$ | 2.1% | $4.23 \times 10^{1}$ ($5.65 \times 10^{2}$) |
| | 256 | TT-RSS ($N = 50$) | $4.12 \times 10^{-1}$ | 1.2% | $1.49 \times 10^{1}$ |
| | | TT-RSS ($N = 100$) | $4.52 \times 10^{-1}$ | 5% | $4.96 \times 10^{1}$ |
| | | TT-CI | $4.39 \times 10^{-1}$ | 1.5% | $8.62 \times 10^{1}$ ($1.70 \times 10^{3}$) |
| | 400 | TT-RSS ($N = 50$) | $5.08 \times 10^{-1}$ | 13.7% | $2.54 \times 10^{1}$ |
| | | TT-RSS ($N = 100$) | $5.09 \times 10^{-1}$ | 15.1% | $8.69 \times 10^{1}$ |
| | | TT-CI | $5.07 \times 10^{-1}$ | 3.9% | $1.55 \times 10^{2}$ ($4.35 \times 10^{3}$) |
| MNIST | 144 | TT-RSS ($N = 50$) | $1.17 \times 10^{1}$ | 31.2% | $8.26 \times 10^{0}$ |
| | | TT-RSS ($N = 100$) | $5.51 \times 10^{0}$ | 20.7% | $2.44 \times 10^{1}$ |
| | | TT-CI | $1.20 \times 10^{0}$ | 84.1% | $2.12 \times 10^{2}$ ($7.37 \times 10^{2}$) |

errors on output values does not directly imply overfitting in the usual sense. It simply indicates that the TT reproduces the output probabilities given by the NN in those cases. However, as we observe in the experiments, and as presented in the experiments of Section 4.1 (specifically, in Fig. 5), accuracies on the set of pivots are usually similar to the accuracies on test samples.

Another notable result from the experiments is that the growth in computation time when increasing $N$ seems to be lower than expected according to the complexity analysis in Section 2.4.

### 3.2.3 Comparison with TT-CI

As in the case of the non-densities, we also compare the performance of TT-RSS and TT-CI for the tensorization of NN models. The results, including relative test errors, percentage of differing classifications, and time costs, are shown in Table 2. As before, times in parentheses for TT-CI denote the total time taken by the full algorithm, including preliminary steps. For TT-RSS, the reported time already includes the entire process, consisting of a single left-to-right sweep. In TT-CI, we also fix the ranks in advance to $r = 10$, and set the number of iterations—one left-to-right and one right-to-left sweep—to 1 for the Bars and Stripes dataset, and 5 for MNIST.

For the Bars and Stripes dataset, we observe very similar test errors, with the most notable difference being in time costs. In this case, both methods yield comparable errors and classifi-

cation differences, while TT-RSS is faster, especially when considering the total time of TT-CI. Notably, for $n = 400$, we observe a larger difference in accuracy, with TT-CI performing better. This may indicate that a larger pivot set is needed to improve the accuracy of TT-RSS in this higher-dimensional model.

The most relevant result for our purposes—and the one that most clearly supports our approach—is the significant loss in accuracy observed for MNIST when using TT-CI. The only configuration we consider for MNIST is $n = 144$, the simplest case. Moreover, we allow TT-CI a larger number of iterations to enable a more extensive search over the model's domain in order to find suitable pivots. Nevertheless, the results show a test error close to 1, suggesting that the TT approximation produced by TT-CI may output values close to 0 across all test points. This is reflected in a classification difference of 84.1%, indicating a completely erroneous model. In contrast, TT-RSS with $N = 100$ yields larger test errors, but a classification difference of only 20.7%, implying that the model has around 80% accuracy. These results highlight how, for more intricate distributions—typically flat with narrow peaks—the `maxvol` routine fails to identify a suitable set of pivots that adequately captures the model's behavior. In such cases, incorporating additional knowledge about the model's domain becomes crucial, as already hinted in Ref. [77].

However, although TT-RSS performs significantly better, its results are still far from optimal. As discussed earlier, training the TT model via gradient descent can achieve accuracies close to the original 98%. Yet this leads to a different model, providing a distinct approximation of the underlying distribution. Our goal through tensorization is to preserve the form of the original NN approximation within a TT representation. This may lead to larger errors, which in turn affect accuracy. This limitation is especially pronounced for MNIST, whose 2D structure, and the spatial patterns likely learned by the original NN, do not align well with the TT format. In contrast, other models, such as the audio classification model discussed in Section 4.1, do not show such drops in accuracy (see Fig. 4).

Finally, we note that using double precision in these examples does not affect accuracy. Only in cases where the function has an exact TT representation, such as in the random TT examples, do errors reach machine precision, and double precision can enhance results. For all other functions and NN models, where errors remain far from machine precision, there is no observable difference between single and double precision, aside from a noticeable increase in time cost when using double precision. For this reason, we use double precision only in Section 4.2, and single precision elsewhere.

## 3.3 Dependence on hyperparameters

In the previous experiments, it was observed that when repeating the same tensorizations multiple times, the results could vary significantly in some situations. This is due to the inherent randomness introduced in TT-RSS, including the random selection of the pivots and the random matrices $U_k$. In this section, we tensorize a single NN model multiple times, varying the hyperparameters of the algorithm—$d$, $r$, and $N$—to examine how they affect accuracy and the variability of results.

For the experiment, we tensorize classification models similar to the ones described in Section 3.2, with the objective of classifying voices by gender. The model and the dataset used (CommonVoice [65]) will be described in more detail in the next section, as this setup will be used for the privacy issue discussed in Section 4.1. For this experiment, we consider only a subset of the CommonVoice dataset where the gender variable is binary, taking values 0 and 1 for women and men, respectively, and is balanced. Therefore, the problem consists of binary classification. In all configurations, the number of input variables is constant, $n = 500$, and the output dimension is 2. We use the polynomial embedding for this experiment.

To run the experiment, we start from an initial baseline configuration: $d = 2$, $r = 5$, and

$N = 50$, and vary only one of these hyperparameters at a time. For each variable, we consider the following ranges: $d \in \{2, 3, 4, 5, 6\}$, $r \in \{2, 5, 10, 20, 50\}$ with $c = 0.99$ in all cases, and $N \in \{10, 20, 50, 100, 200\}$. For each configuration, the TT-RSS algorithm is run 50 times, and for each result, we obtain the test accuracy, defined as

$$\text{acc}(p, \mathbf{s}, \mathbf{y}) = \frac{1}{M} \sum_{i=1}^{M} \chi_{\{\arg\max_y p(y|\mathbf{s}^i)\}}(\mathbf{y}^i), \tag{41}$$

where $\chi$ denotes the indicator function, $\mathbf{s}$ is the set of test samples, and $\mathbf{y}$ is the vector of labels for the samples. Results are presented in Fig. 3.

The clearest result observed, which coincides with expectations, is that increasing the number of pivots $N$ improves the approximation, leading to greater accuracies. Most notably, also as expected, the more pivots we use, the less variance the results exhibit.

In contrast, this behavior is not observed with the growth of $d$ and $r$, even though this might not always be the general case. Specifically, the allowed rank $r$ determines the number of singular vectors retained in the TRIMMING stage (recall Section 2.2.3), which helps reduce variance arising from the small amount (relative to the exponential number of possibilities) of randomly chosen pivots. Therefore, allowing greater values of $r$ may lead to error accumulation. However, the greater the number of pivots $N$, the less variance there will be, and greater ranks $r$ could be allowed.

Moreover, for Eq. (21) to have an exact solution, $B_k(\beta_{k-1}, (i_k, \alpha_k))$ should be in the span of the column space of matrix $A_{k-1}(\beta_{k-1}, \alpha_{k-1})$. In cases where $N > r$, and considering that both $A_{k-1}$ and $B_k$ are already linear approximations as solutions of Eq. (23), it is likely that the columns of $B_k$ are not in the span of the column space of $A_{k-1}$. This becomes even more unlikely as $d$ increases. More importantly, since the TT is formed by the contraction of all the cores $G_k$, these approximation errors may accumulate rapidly, leading to steep decreases in accuracy with increasing $d$ and $r$, as observed in the left and center plots of Fig. 3.

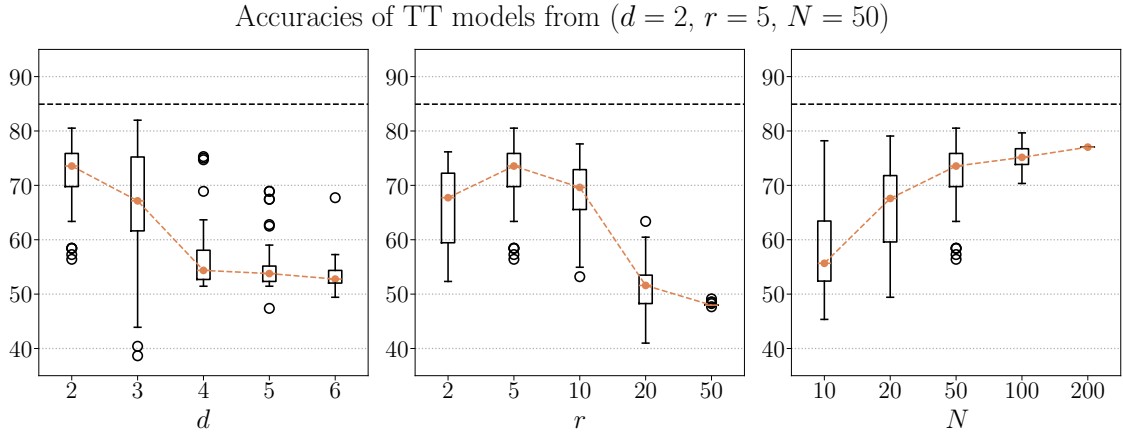

Figure 3: Distribution of the accuracies of 50 tensorized models for each configuration, represented using box plots. Each configuration is made by varying one of the hyperparameters $d$, $r$, and $N$ from the baseline point $d = 2$, $r = 5$, $N = 50$. The orange lines connect the medians across all the different values for a hyperparameter. The horizontal dashed black lines represent the original accuracy of the NN model being tensorized.

# 4 Applications: Privacy and interpretability

In this section, we present the two main results of our work, involving the application of TT-RSS in more realistic scenarios to fulfill different purposes. First, in Section 4.1, we expand the privacy experiments presented in Ref. [30] to the case of classification models for speech datasets. We show that, by training via common gradient descent methods, an attacker with access to the parameters of an NN model might be able to learn properties of the training dataset, which poses a privacy concern. To address this, we propose tensorizing the model with TT-RSS and obfuscating the training information by leveraging the well-characterized gauge freedom of TT models. Secondly, in Section 4.2, we use TT-RSS to reconstruct the TT representation of the AKLT state from black-box access to the amplitudes of a limited number of spin configurations. From this representation, we calculate an order parameter that allows us to estimate the SPT phase in a similar way to the approach presented in Ref. [68]. While this serves as a proof of principle, the methodology could be extended to estimate SPT phases in higher-dimensional settings.

## 4.1 Privacy

According to a previous work [30], a privacy vulnerability has been observed that might affect machine learning models trained via gradient descent methods. This is typically the case for NN models, whose intricate architectures can express a wide variety of functions but may also memorize undesired information, thereby raising privacy concerns. In this vulnerability, it was shown that during the learning process of several models, the distribution of the parameters began to exhibit biases associated with biases in certain features of the dataset. More significantly, this effect was observed even when the biases appeared in features that were not relevant to the learning task. To address this issue, it was proposed to replace NN models with TNs, whose gauge freedom is well-characterized. This allows the parameters of the models to be redefined independently of the learning process, effectively removing hidden patterns.

The experiments conducted in Ref. [30] were set in a relatively simplistic scenario. The dataset consisted of only five features, plus an additional irrelevant variable introduced to perform the attacks. It could be argued that, in more realistic scenarios, such an irrelevant feature would likely be removed during an initial pre-processing stage, typically involving techniques like Principal Component Analysis to reduce dimensionality and select relevant features.

To address this limitation, we present a much more realistic scenario in this section, where the objective is to classify voices by gender. In this case, each voice sample is represented as a vector in $[0, 1]^{500}$, and the irrelevant feature targeted for the attack is one that would not typically be removed during standard pre-processing and does not correspond to a single input variable: the accent. To demonstrate the privacy leakage, we train NN models via gradient descent using training datasets where the accents are predominantly of one type or another. As a solution, we show that instead of training TT models from scratch, TT-RSS can be used to tensorize the original models, reconstructing models that are private while maintaining nearly the same accuracies. In cases where the accuracies drop, it is possible to re-train the TT models for a few additional epochs. In either case, this approach is more efficient and straightforward than training new TT models from the ground.

### 4.1.1 Datasets

For this experiment, we use the public dataset CommonVoice [65], which provides a collection of speech audios in mp3 format. Each audio sample is associated with demographic information, including gender, age, and accent. Additionally, each audio is tagged with a unique

identifier, `client_id`, corresponding to each speaker. To simplify the task, we train classifiers with the goal of differentiating voices by gender. To reduce the problem to binary classification, we keep only voices labeled as either woman or man. The variable targeted by the attacker is the accent. To simplify this variable to a binary classification, we consider only "England English" and "Canadian English" accents, referred to here simply as "English" and "Canadian".

To enable statistical analysis, we partition the dataset into several subsets, training multiple NNs on the different subsets. The partitions are also stratified based on the proportion of the `accent` variable. First, we extract a subset from the predefined `train` partition of CommonVoice and balance the `gender` variable across the accents. Denoting $q$ as the percentage of English speakers, we form, for each $q \in \{1, 5, 10, 20, 30, 40, 50, 60, 70, 80, 90, 95, 99\}$, 10 datasets. For each value of $q$, we create an imbalanced dataset according to the specified accent proportion, and sample 80% of the `client_id` identifiers to form the 10 datasets, splitting each of them into training and validation subsets. Consequently, the 10 datasets created for each $q$ value are distinct but not disjoint. From the remaining 20% of IDs in each case, we select around 200 samples to serve as the pool from which the $N$ pivots for tensorizing the NN models are chosen. Thus, the pivots used in TT-RSS exhibit the same imbalances as those present in the corresponding training datasets. These subsets of 200 samples will be used also to re-train the TT models.

For pre-processing the audio samples, we first resample all audios to a frequency of 1 000 Hz and crop each to retain only the middle 1 000 values, corresponding to one second of audio per person. Next, we apply the Fast Fourier Transform (FFT) and retain only half of the values, since the FFT of real-valued functions is Hermitian-symmetric. This results in feature vectors with 500 variables. Finally, we renormalize the values to restrict them to the interval $[0, 1]$.

### 4.1.2 Models

For the classifiers, we use shallow fully connected neural networks consisting of a single hidden layer with 50 neurons and an output layer with 2 neurons (corresponding to the 2 classes), followed by a softmax layer. Between these layers, we include dropout and ReLU activation functions.

The models are trained using the Adam optimizer to minimize the negative log-likelihood. To determine appropriate hyperparameters, we perform an optimization procedure to tune the hyperparameters by training 100 models for 5 epochs each, using balanced datasets ($q = 50$). The optimal hyperparameters obtained are: dropout rate = 0.1, batch size = 8, learning rate = $3.6 \times 10^{-3}$, and weight decay = $2.6 \times 10^{-5}$.

Using these hyperparameters, we train 25 models for 20 epochs on each of the 10 datasets corresponding to different imbalance levels $q$. The resulting models achieve an average accuracy of approximately 82%.

### 4.1.3 Tensorization

To ensure a realistic tensorization scenario, we consider having only black-box access to the NN model, meaning that we can only observe its classifications. While this might seem restrictive, it establishes a minimal-information scenario. For model providers with full access to their models, tensorization could use the output probabilities for each class instead of just the predicted class. However, in other cases, such as when the individual interested in tensorizing the model is merely a user, this black-box approach applies. Additionally, for tensorization, we select as pivots points that are not part of the training or validation sets. Thus, we assume knowledge of the training dataset's distribution but no direct access to the dataset itself.

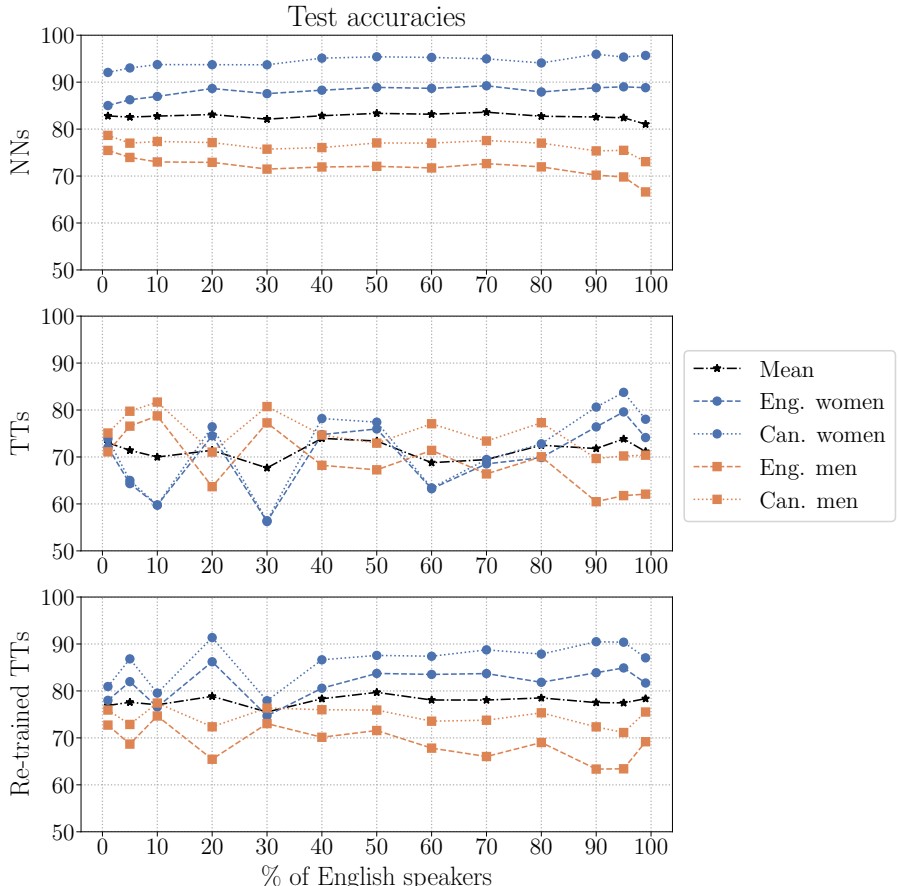

Figure 4: Accuracy of the different models evaluated on a test dataset consisting of 300 points, equally distributed across the 4 subgroups: English woman, Canadian woman, English man, and Canadian man. From top to bottom, the models are: the original NN models, TT models output by TT-RSS, and TT models re-trained for 10 epochs. Accuracies are measured for all percentages $q$ of English speakers present in the datasets.

*Remark* 4.1. Tensorizing models with only black-box access can be considered a form of model-stealing attack, as it provides a method to reconstruct models using only user-level access [114].

The function to be tensorized is thus defined as:

$$\hat{q}_{\text{NN}}(x, y) := \begin{cases} 1, & \text{if } \hat{p}_{\text{NN}}(x, y) \geq \frac{1}{2}, \\ 0, & \text{otherwise.} \end{cases} \tag{42}$$

For TT-RSS, we choose hyperparameters that balance accuracy and computational cost, as suggested by the results in Section 3.3. Specifically, we set $d = 2$, $r = 5$ with $c = 0.99$, and $N = 100$, always using the polynomial embedding.

Tensorized models initially exhibit reduced accuracies, typically around 70%-75%. To address this, we re-train the models for 10 epochs using a small dataset containing the pivots but disjoint from the training and validation sets. This process improves accuracy to approximately 78%-80%. While this represents a slight drop compared to the original NN models, it is an acceptable tradeoff for enhanced privacy. Figure 4 compares the accuracies of the original NN models to the tensorized and re-trained models.

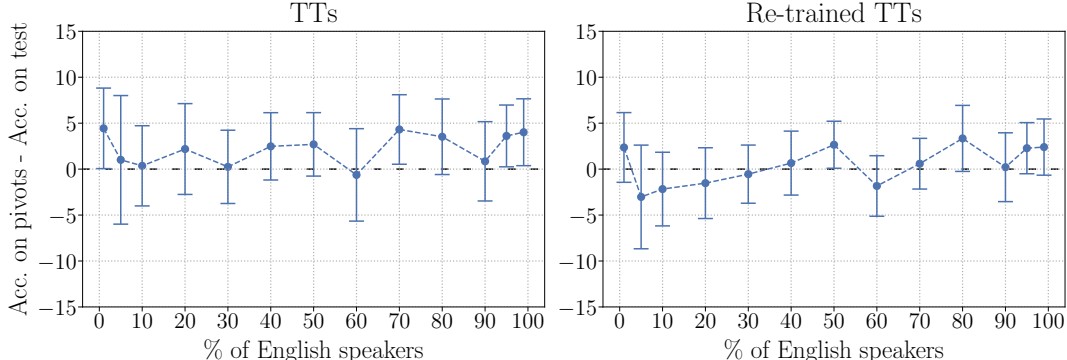

Figure 5: Difference between the accuracy measured only on the pivots used in TT-RSS and the accuracy on test samples. On the left, accuracies are measured for TT models directly output from the TT-RSS decomposition. On the right, accuracies are for TT models after re-training them for 10 epochs on a small dataset containing the pivots. These differences are measured for all percentages $q$ of the presence of English speakers in the datasets. For each $q$, mean differences in accuracy across all models trained for that $q$ are shown, with error bars at $\pm 0.5\sigma$, where $\sigma$ denotes the standard deviation.

Previously, we discussed potential issues arising from overfitting the pivots in TT-RSS. As highlighted in Section 3.2.2, overfitting the NN model's outputs does not necessarily imply overfitting of the TT model to the pivots, evidenced by higher accuracies on these points compared to other unseen samples. In our experiments, we verify this by comparing accuracies of TT models on pivots and test samples. Although both plain TT models from TT-RSS and re-trained models show minimal accuracy differences, there is a slight bias towards higher accuracy for pivots in the plain TT models. However, this behavior becomes unclear after re-training (see Fig. 5).

Upon re-training, we proceed to obfuscate the TT models' parameters to eliminate any lingering patterns. The approach proposed in Ref. [30] involves using the univocal canonical form of the TT. This method provides new TT cores based on a small number of evaluations, ensuring that the TT representation contains the same information as the black-box model. However, as demonstrated in Fig. 6, if exploitable patterns exist in the black-box model, they will persist in the TT cores. To mitigate this, we propose a different approach.

To obfuscate the TT cores, we introduce random orthogonal (in general, unitary) matrices $W_k$ and their transposes (equivalently, adjoints), $W_k^\top$, between contractions. This results in new cores defined as

$$\widetilde{G}_k(\alpha_{k-1}, x_k, \alpha_k) := W_{k-1}^\top(\alpha_{k-1}, \theta_{k-1}) G_k(\theta_{k-1}, x_k, \theta_k) W_k(\theta_k, \alpha_k). \tag{43}$$

Since the added matrices are orthogonal, they cancel out with their transposes in adjacent contractions, preserving the TT's overall behavior. Choosing $W_k$ as random orthogonal matrices introduces randomness into the cores $\widetilde{G}_k$, effectively removing hidden patterns while maintaining stability by preserving core norms. This new representation is referred to as Private-TT.

### 4.1.4 Attacks

The property we aim to infer from the training dataset is the majority accent present in it. Specifically, we aim to create a function that, given some information about a model, predicts the majority accent. To achieve this, we train a logistic regression model.

As detailed earlier, we trained 250 NNs for each percentage $q$. To train the attack model, we merge all models corresponding to $q$ and $100-q$, for $q \in \{1, 5, 10, 20, 30, 40\}$, and create a dataset by storing some information from each model, $x$, along with its corresponding label, $y$. We define the label for each model as $y = 0$ if $q < 50$, and $y = 1$ otherwise. For the information representing the model, $x$, we consider two scenarios: (i) *black-box* access, where we can only observe classifications made by the model, and (ii) *white-box* access, where we have full access to the model's parameters.

For the black-box scenario, we evaluate each model on 100 samples, equally divided among four groups: woman or man, English or Canadian speakers. The resulting 100-dimensional vector represents the information we observe from each model. To ensure fairness, the same 100 samples are used to evaluate all models. These samples are selected from a predefined partition of the CommonVoice dataset, denoted as `validated`, ensuring no overlap with the `train` partition to avoid including training points from any model.

For the white-box scenario, we assume complete access to the model parameters. Although an attacker with white-box access can also evaluate the model (thus accessing black-box information), we conduct these experiments separately to assess the performance of attacks based solely on one type of information. Stronger attacks combining both information sources are possible but beyond the scope of this analysis.

To compare attack accuracies, we create such datasets for three model types: NNs, TTs, and Private-TTs. Unlike what was observed in Ref. [30], our results reveal that accent influences the models' gender classification even at a black-box level (see Fig. 6). Models trained with a majority of English speakers can be distinguished from those trained on datasets dominated by Canadian speakers based on their performance differences on individuals with these accents.

Additionally, although TT models are constructed via TT-RSS using only black-box information—implying that NNs and TTs allegedly represent the same black-boxes—black-box attacks on TT models perform worse than on NNs. This discrepancy could be attributed to the rank restriction $r$ of the TT, which limits the model's expressive power and thereby reduces its tendency to memorize training data. Consequently, this constraint slightly lowers the accuracy of the TT models, as shown in Fig. 4. Notably, black-box attacks on Private-TT models yield identical results to those on TT models, as the black-box outputs remain the same.

The most revealing results arise from white-box attacks, as they demonstrate the expected behavior for each case. For NN models, a simple logistic regression attack achieves nearly 100% accuracy across all imbalance percentages. For TT models, attack accuracy increases with greater imbalance, mirroring the trend observed in black-box attacks. This similarity arises because TT-RSS reconstructs the TT using only black-box evaluations of the model. Thus, white-box attacks on TT models produce results comparable to black-box attacks. As discussed in Ref. [30], this could already be considered private to some extent. However, it remains unclear whether the information in the black-box and white-box representations is equivalent or complementary. An attacker might achieve better results by combining both. To mitigate such risks, all information in the white-box can be obfuscated by applying the proposed obfuscation method for TTs, which is computationally inexpensive. This is reflected in the attacks on Private-TTs, where accuracy drops to 50%, equivalent to random guessing.

It is important to note, however, that these attacks rely on simple logistic regression models and should be considered lower bounds for what an attacker could achieve in this setting. On the other hand, the attacks assume access to extensive information, such as hundreds of trained models, enabling statistical analysis and training of powerful classifiers. From this perspective, the results in Fig. 6 may represent a pessimistic lower bound on the potential privacy risks.

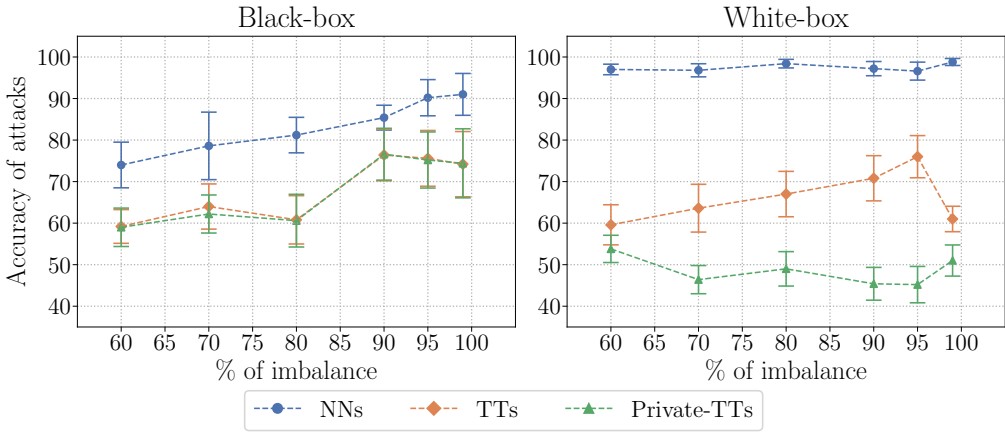

Figure 6: Accuracy of logistic regression models trained to classify whether a model was trained on a dataset with a majority of English speakers or a majority of Canadian speakers. For each percentage of imbalance, a dataset is formed from 500 models trained on 20 different datasets (10 datasets per percentage $q$, with 25 models per dataset). For each configuration, attacks are evaluated via $K$-fold validation with $K = 10$, and mean accuracies are displayed with error bars at $\pm 0.5\sigma$, where $\sigma$ denotes the standard deviation.

## 4.2 Interpretability

Tensor networks have recently been proposed as machine learning models that could be more interpretable than NNs, as they provide direct access to meaningful information about the underlying distribution [51,52]. However, their initial acclaim stemmed from their ability to study properties of quantum many-body states, both theoretically and practically. Specifically, TN representations have facilitated the study of topological phases of matter, which are known to be in one to one correspondence with the symmetries present in the tensors of the network [33]. This approach has been particularly successful for the case of SPT phases, which can be classified by examining how global symmetries of the state affect the local representation of the tensors in the network [101–107].

In recent years, several approaches have emerged to approximate quantum states via neural network representations, enabling the efficient representation of complex systems [13–17]. However, these representations often lack the capacity to analyze properties of the states directly, requiring additional models to estimate each property separately [18–20]. In contrast, the methodology we propose leverages pre-trained NNQS to derive a TN representation, enabling the direct study of such properties.

### 4.2.1 AKLT model

We present this methodology in a simplified scenario that may serve as a proof of principle. In particular, we will estimate the SPT phase of the AKLT state [67], which admits a TT representation that was already derived in early works [115,116]. Although this state admits a TT form, this is not necessary for the method to succeed, since we will only allow black-box access to this function. Indeed, we will exploit the TT structure of the state only when obtaining the pivots and only for computational efficiency. Thus, this setup is analogous to having access to other less interpretable models, such as a NN, which we can query with spin configurations to obtain the associated amplitudes. Using this function, we will reconstruct a new TT representation of the state via TT-RSS and use this representation to compute the order parameter defined in Ref. [68].

The AKLT model is defined by the following Hamiltonian:

$$H = \sum_{k=1}^{n} P_{k,k+1}^{(2)}, \tag{44}$$

where $P_{k,k+1}^{(2)}$ is the projector operator that projects the total spin of the neighboring sites $k$ and $k+1$, both of them spin-1 particles, onto the spin-2 subspace. Consequently, its ground state, referred to as the AKLT state, is formed by spin-1/2 singlets between nearest neighbor sites, with projector operators at each site projecting the two spin-1/2's into the spin-1 subspace.

This state can be represented in TT/MPS form. For the representation of the model and the subsequent description of the order parameter computation, we use Dirac's bra-ket notation. Additionally, instead of representing the TT cores as tensors $G_k(\alpha_{k-1}, x_k, \alpha_k)$, we will simply denote them as matrices indexed by the input label, $G_k^{x_k}$. Thus, a translationally invariant TT representation of the AKLT state is defined as:

$$|\Psi\rangle = \sum_{x_1,\dots,x_n} \text{Tr}[G^{x_1}\cdots G^{x_n}]|x_1\cdots x_n\rangle, \tag{45}$$

where subscripts are omitted as all cores are identical. The cores are given by

$$G^- = -\sqrt{\frac{2}{3}}\sigma_-, \qquad G^0 = -\sqrt{\frac{1}{3}}\sigma_0, \qquad G^+ = \sqrt{\frac{2}{3}}\sigma_+. \tag{46}$$

After a suitable change of basis, these tensors become

$$G^{x_k} = \sqrt{\frac{1}{3}}\sigma_{x_k}, \quad x_k \in \{1,2,3\}, \tag{47}$$

as detailed in Example 1.e of the Appendix of Ref. [33]. For clarity in the presentation of Section 4.2.3, we assume the TT cores are expressed in this basis.

In the above definitions, $\sigma$ denotes the Pauli matrices, defined as:

$$\sigma_1 = \begin{pmatrix} 0 & 1 \\ 1 & 0 \end{pmatrix}, \qquad \sigma_2 = \begin{pmatrix} 0 & -i \\ i & 0 \end{pmatrix}, \qquad \sigma_3 = \begin{pmatrix} 1 & 0 \\ 0 & -1 \end{pmatrix}. \tag{48}$$

The matrices $\sigma_-$, $\sigma_0$, and $\sigma_+$ are then defined as:

$$\sigma_- = \sigma_1 - i\sigma_2, \qquad \sigma_0 = \sigma_3, \qquad \sigma_+ = \sigma_1 + i\sigma_2. \tag{49}$$

*Remark* 4.2. Until now, we have focused on applying TT-RSS to real-valued functions. However, it is important to note that the tensorization process is equally applicable to complex-valued functions, as it involves only linear projectors, SVDs, and the solving of matrix equations. Consequently, while in this example we use TT-RSS to recover a TT representation of the AKLT state, which has real coefficients, the methodology can be straightforwardly extended to general quantum states with complex coefficients.

### 4.2.2 Tensorization

The definition given in Eq. (45) forms a TT with periodic boundary conditions (PBC). However, TT-RSS cannot generate TTs with this type of loop. Therefore, the function we decompose is defined by imposing open boundary conditions (OBC), as follows:

$$|\Psi\rangle = \sum_{x_1,\dots,x_n} \langle 0| G^{x_1}\cdots G^{x_n} |0\rangle |x_1\cdots x_n\rangle, \tag{50}$$

from which we can define $G_1^{x_1} = \langle 0| G^{x_1}$ and $G_n^{x_n} = G^{x_n} |0\rangle$.

To apply TT-RSS, we fix the input dimension and rank as defined by the AKLT state, i.e., $d = 3$ and $r = 2$ (with $c = 1 - 10^{-5}$). Since the function has a discrete domain, we use the trivial embedding for all sites. For pivot selection, we assume no knowledge of the state, and thus, we sample random configurations $(x_1, \ldots, x_n)$ from the distribution defined by the AKLT state. While we assume no access to the TT representation of the state during the entire process, we leverage it for computational efficiency when obtaining the pivots, as it enables direct sampling. Alternatively, this step could be performed via Monte Carlo sampling, as required for NNQS. To analyze the performance of TT-RSS with different numbers of pivots, we use values of $N$ ranging from 2 to 14. Additionally, we run the experiments for three different values of $n$: 100, 200, and 500. For each configuration, TT-RSS is applied 10 times.

Upon tensorization, we obtain approximate versions of $|\Psi\rangle$, given by

$$|\widehat{\Psi}\rangle = \sum_{x_1, \ldots, x_n} \widehat{G}_1^{x_1} \cdots \widehat{G}_n^{x_n} |x_1 \cdots x_n\rangle . \tag{51}$$

Note that in the reconstructed TT the cores may be different, as no restrictions are imposed in Eq. (9) to obtain equal cores. To attempt a TI representation, we can bring the TT into a canonical form [42, 117]. In this case, we only fix the gauge to the left of each tensor by requiring

$$\sum_{x_k} \widehat{G}_k^{x_k \dagger} \widehat{G}_k^{x_k} = \mathbb{I} . \tag{52}$$

This condition can be achieved iteratively through SVD, as described in Ref. [42], resulting in what we refer to as the SVD-based canonical form. In this form, the representation of the AKLT state obtained is equivalent to the one in Eq. (47) up to conjugation by unitaries, i.e.,

$$\widehat{G}_k^{x_k} = W_{k-1}^\dagger G^{x_k} W_k , \tag{53}$$

where $W_k \in \mathrm{U}(2)$ for all $k \in [n-1]$.

To evaluate the accuracy of the reconstruction, we measure the fidelity between the recovered state $|\widehat{\Psi}\rangle$ and the original state $|\Psi\rangle$, as defined in Eq. (37). Remarkably, we observe that using as few as 12 pivots is sufficient to achieve a fidelity of 1, meaning that the AKLT state is recovered exactly, regardless of the system size $n$ (see Fig. 7).

### 4.2.3 Order parameter

To estimate the SPT phase of the AKLT state, we compute the order parameter described in Ref. [68], which distinguishes the equivalence classes of the projective representations of the symmetry group $\mathbb{Z}_2 \times \mathbb{Z}_2$. This order parameter can be computed using the TT representation of the state and is insensitive to unitary conjugations of the form of Eq. (53). As such, it provides a natural method for estimating SPT phases within our approach.

Specifically, if $u_g$ denotes a unitary representation of $\mathbb{Z}_2 \times \mathbb{Z}_2$, due to the symmetry, the system satisfies

$$u_g^{\otimes n} |\Psi\rangle = |\Psi\rangle . \tag{54}$$

This implies, by virtue of the Fundamental Theorem of MPS [42], that there must exist a TI representation of $u_g^{\otimes n} |\Psi\rangle$ given by tensors $H$, such that

$$H^{x_k} = V_g^\dagger G^{x_k} V_g , \tag{55}$$

where the physical transformation $u_g$ induces a transformation on the virtual degrees of freedom, represented by $V_g$. These transformations are not unique; transformations of the

form $\widetilde{V}_g = \beta(g)V_g$, with $\beta(g) \in U(1)$, also satisfy Eq. (55). Consequently, they yield projective representations of the symmetry group, which can be classified into equivalence classes. These equivalence classes form a group isomorphic to the second cohomology group, $H^2(\mathbb{Z}_2 \times \mathbb{Z}_2, U(1))$, which labels the inequivalent representations.

For the AKLT state with global on-site $\mathbb{Z}_2 \times \mathbb{Z}_2$ symmetry, we aim to compute an order parameter to distinguish whether the transformations $V_g$ and $V_h$, corresponding to different group elements $g$ and $h$, commute or anti-commute. In this example, it is straightforward to verify, using the TI representation of the AKLT state (47), that the Pauli matrices, which anti-commute, form a projective representation of $\mathbb{Z}_2 \times \mathbb{Z}_2$. The relationship between the elements of the symmetry group and their corresponding representations as Pauli matrices is shown in Table 3.

However, since the TT representation returned by TT-RSS is not TI and retains the gauge freedom described in Eq. (53), projective representations may not be readily identifiable. To address this, we employ the order parameter proposed in [68], which determines whether the matrices of the projective representation commute or anti-commute without requiring direct access to their explicit forms.

Due to the finite correlation length of TTs, it can be shown that if $U_g$ denotes the action of the symmetry under renormalization (e.g., blocking $L$ sites, where $U_g = u_g^{\otimes L}$), then

$$G^{x_k} \cdots G^{x_{k+L}} U_g(x_k, \ldots, x_{k+L}, y_k, \ldots, y_{k+L}) \bar{G}^{y_k} \cdots \bar{G}^{y_{k+L}} \approx (\Lambda V_g) \otimes (\Lambda \bar{V}_g), \qquad (56)$$

where $\Lambda = \sum_i \lambda_i |i\rangle \langle i|$, with $\lambda_i = \frac{1}{\sqrt{2}}$ for $i \in [2]$ being the Schmidt values. This approximation converges exponentially fast with the blocking size $L$. For the subscript notation, we use the Pauli matrix index $i$ to label the different group elements and their corresponding representations, as shown in Table 3, that is, $g_i \leftrightarrow u_i \leftrightarrow V_i \leftrightarrow \sigma_i$.

Using this, we can compute the following order parameter:

$$\mathcal{E}_L(\Psi) := \langle \Psi | (U_3 \otimes U_3 \otimes \mathbb{I}) \mathbb{F}_{13} (U_1 \otimes U_1 \otimes \mathbb{I}) | \Psi \rangle, \qquad (57)$$

where $\mathbb{F}_{13}$ is the operator that swaps the first and third sites. This value effectively computes

$$\mathcal{E}_L(\Psi) \approx \text{Tr}[V_3 V_1 V_3^\dagger V_1^\dagger \Lambda^4] \text{Tr}[\Lambda^4], \qquad (58)$$

yielding

$$\mathcal{E}(\Psi) \approx \pm \text{Tr}[\Lambda^4]^2, \qquad (59)$$

where the sign reveals whether the elements $V_1$ and $V_3$ commute or anti-commute.

Table 3: Correspondence between elements $g_i$ of the symmetry group $\mathbb{Z}_2 \times \mathbb{Z}_2$, elements $u_i$ of a unitary representation, and elements $V_i$ of a projective representation.

| Element of $\mathbb{Z}_2 \times \mathbb{Z}_2$ | Unitary rep. | Projective rep. |
|---|---|---|
| $g_3 = (1,0)$ | $u_3 = \begin{pmatrix} -1 & 0 & 0 \\ 0 & -1 & 0 \\ 0 & 0 & 1 \end{pmatrix}$ | $V_3 = \sigma_3$ |
| $g_2 = (0,1)$ | $u_2 = \begin{pmatrix} -1 & 0 & 0 \\ 0 & 1 & 0 \\ 0 & 0 & -1 \end{pmatrix}$ | $V_2 = \sigma_2$ |
| $g_1 = (1,1)$ | $u_1 = \begin{pmatrix} 1 & 0 & 0 \\ 0 & -1 & 0 \\ 0 & 0 & -1 \end{pmatrix}$ | $V_1 = \sigma_1$ |

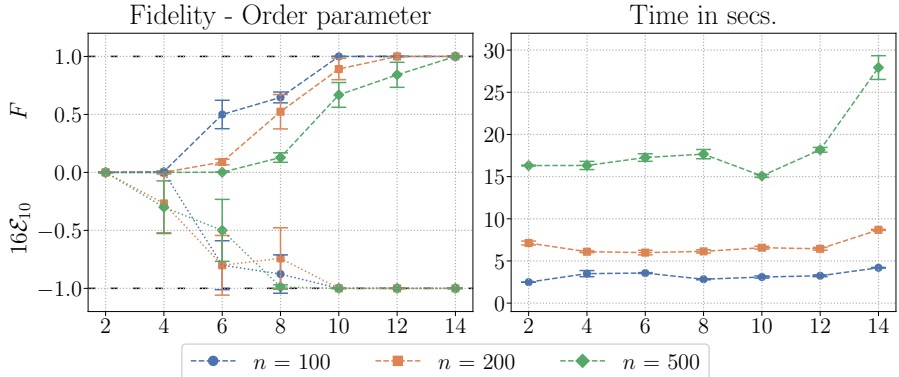

Figure 7: Fidelity between the AKLT state $|\Psi\rangle$ and the approximated TT representation obtained via TT-RSS, $|\widehat{\Psi}\rangle$, along with the order parameter $16\mathcal{E}_{10}$ to distinguish the trivial from the non-trivial phase. The right figure shows the time taken to perform TT-RSS in seconds. For each value of $N$, the decomposition is performed 10 times, displaying the mean values with error bars at $\pm 0.5\sigma$, where $\sigma$ denotes the standard deviation. Three configurations are considered with different numbers of variables, $n = 100$, $n = 200$, and $n = 500$, represented by different colors. All computations are performed in double precision.

*Remark* 4.3. In the description of the order parameter $\mathcal{E}_L$, we assume that the TT is TI. However, as pointed out earlier, the result from TT-RSS after canonicalization still retains a U(2) gauge freedom. Nevertheless, when computing the order parameter while accounting for the unitary transformations $W_k$ acting on the virtual indices as shown in Eq. (53), these transformations cancel out, yielding the same result as in the TI case.

In our experiment, we measure $16\mathcal{E}_{10}(\Psi) \approx \pm 1$ to distinguish the trivial from the nontrivial phase. For the AKLT state, which is known to be in the non-trivial phase due to the anti-commutation relations of the Pauli matrices, the order parameter is $-1$. This is the value we must observe from the reconstructed TT representation of the state as well. As shown in Fig. 7, when the fidelity is 1—indicating an exact reconstruction of the AKLT state—the order parameter attains the value of $-1$. However, since our method does not rely on prior knowledge of the exact representation, it is applicable to cases where the achieved fidelity is less than 1. Notably, our results demonstrate that the order parameter reaches $-1$ with only 6 pivots in TT-RSS, even when fidelity remains below 1.

This demonstrates that our method can estimate order parameters of systems with hundreds of sites in just a few seconds, in sharp contrast to other machine learning techniques that require training new models to learn specific system properties [18–20]. With our approach, one can leverage black-box access to a pre-trained model, such as a NNQS [13–17], to construct a TN representation via TT-RSS and use it to study system properties with well-established tools in the field.

For systems well-approximated by TTs, as in the example shown, it may be more beneficial to perform variational optimization directly using the TT form, which can be achieved very efficiently [118,119]. However, for systems in 2D or higher dimensions, this approach becomes highly complex, as it has been shown that contracting Projected Entangled Pair States (PEPS) is #P-complete [120,121]. If TT-RSS could be extended to such higher-dimensional scenarios, this methodology could be used to construct PEPS while avoiding their optimization. Although no current adaptation exists for PEPS, ideas involving sketching and cross interpolation have already been employed to construct Tree TNs [78,86,87], which could be implemented for higher-dimensional layouts.

# 5 Further results: Initialization and compression

As a side effect of tensorizing machine learning models via TT-RSS for privacy and interpretability, we observe two additional phenomena that could be independently exploited and deserve further analysis.

On the one hand, following the methodology outlined in Section 4.1, if the TT obtained through TT-RSS exhibits lower accuracies compared to the original NN models, these TTs could still serve as starting points for further optimization. This suggests that TT-RSS could be utilized as an initialization mechanism for TT machine learning models. We explore this potential by comparing different initializations (corresponding to different embeddings) in Section 5.1.

On the other hand, TT-RSS allows us to leverage one of the most significant advantages of tensor networks: their efficiency. By tensorizing NN models, we can produce TT models characterized by fewer parameters, leading to reduced computational costs. In Section 5.2, we compare the compression capabilities of this approach with other common techniques that tensorize NNs in a layer-wise fashion to reduce parameters, showing that constructing a single TT model with TT-RSS might offer superior memory-time efficiency trade-offs.

## 5.1 Initialization

When training machine learning models via gradient descent methods, the initialization of variational parameters is crucial for ensuring stable optimization and achieving high-quality solutions. For DNNs with ReLU activations, it has been demonstrated that initializing weights from a Gaussian distribution with appropriate standard deviations results in the outputs of the initial network being distributed according to the Gaussian density $\mathcal{N}(0,1)$ [122]. Furthermore, over-parameterized DNNs with such random initialization have been shown to converge to near-global minima in polynomial time when optimized using Stochastic Gradient Descent (SGD) [123].

Inspired by the success of these methods, a similar approach was proposed for TNs in Ref. [48], aiming to ensure that the outputs of randomly initialized TNs follow the $\mathcal{N}(0,1)$ distribution. However, the sequential contraction of tensors in a TN involves multiple additions and multiplications of the parameters within the tensors, making the outputs highly sensitive to perturbations. This sensitivity often leads to rapid explosion or vanishing of both the outputs and gradients, which can render optimization infeasible. Consequently, this type of initialization is only effective for shallow networks with relatively few input variables. As a partial remedy for the vanishing or exploding values, recent work has introduced a method to maintain the total norm of the TN within a desired range by appropriately normalizing the cores [49]. In general, if TN models are employed in scenarios where the total norm is not a critical factor, the intermediate computations during contraction can be stabilized by normalizing the results at each step. However, this normalization process is typically performed independently for each input sample, which may undermine the ability to compare the magnitudes of different outputs. In particular, if the TN model represents a probability distribution, it is preferable to use a consistent scaling factor, or partition function, across all samples to preserve the comparability of outputs.

Using TT-RSS, we can generate TT models that produce outputs within a desired range for inputs drawn from a specified distribution and for a chosen set of embedding functions. As an example, we train TT models for the same classification task described in Section 4.1 with various initializations and embeddings, and compare their performance against TT models initialized via TT-RSS.

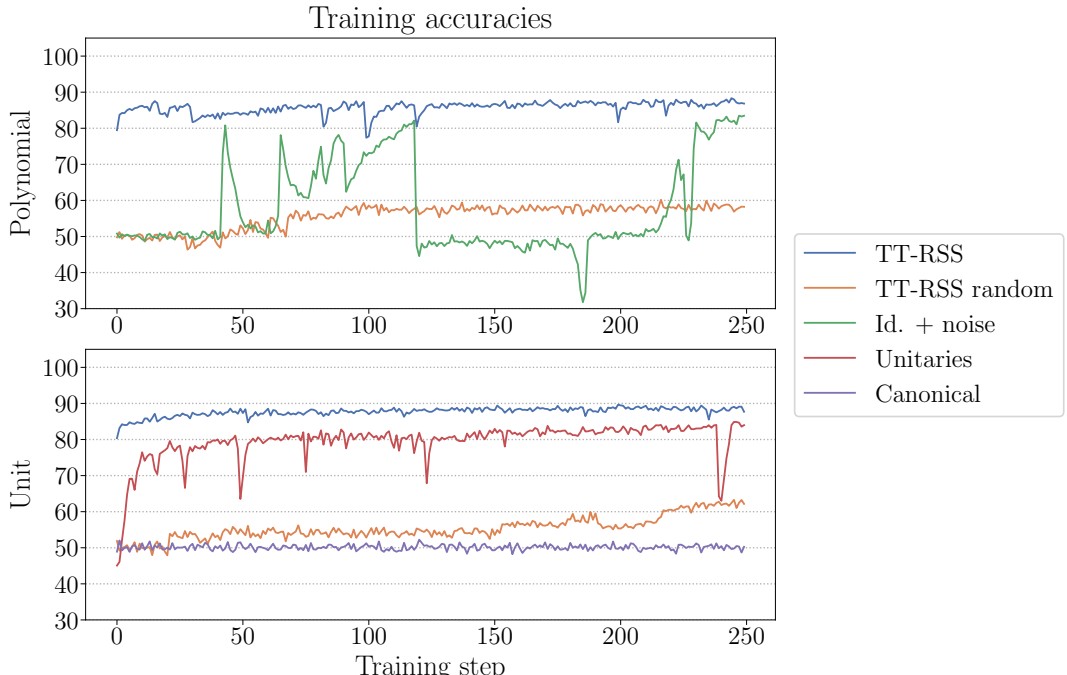

Figure 8: Evolution of training accuracies of TT models trained via the Adam optimizer (learning rate $= 10^{-5}$, weight decay $= 10^{-10}$) for a binary classification task. The top figure shows results using the polynomial embedding, while the bottom figure corresponds to the unit embedding. Different initialization methods are employed for both embeddings, with each method represented by a distinct color.

For the embeddings, we consider two options: (i) the polynomial embedding $\phi(x) = [1, x]$, as proposed in Ref. [41], and (ii) the *unit* embedding

$$\phi(x) = \left[ \cos\left(\frac{\pi}{2}x\right), \sin\left(\frac{\pi}{2}x\right) \right],$$

introduced in Ref. [40] and commonly used for machine learning tasks. In all cases, we set $d = 2$ and $r = 5$. For the initializations, we explore the following settings:

- **TT-RSS:** The TT model is obtained by tensorizing a pre-trained NN with the architecture described in Section 4.1.2. This initialization provides an initial configuration of ranks $r_1, \ldots, r_{n-1}$ adjusted to the given distribution. For all other initializations, the rank $r$ is fixed across all internal dimensions.

- **TT-RSS random:** The TT is generated via TT-RSS from a random function $p$ that outputs random probabilities, only ensuring that the conditional probabilities $p(y|x)$ sum to 1. Consistent with prior cases, the function to be tensorized is the square root of $p$.

- **Id. + noise:** The first matrix in each core is initialized as the identity, i.e., $G_k^1 = \mathbb{I}$. Gaussian noise with a small standard deviation (ranging from $10^{-5}$ to $10^{-9}$) is then added to the entire tensor, ensuring that outputs remain stable. This initialization, combined with the polynomial embedding, was suggested in Ref. [41].

- **Unitaries:** Each core is initialized randomly as a stack of orthogonal matrices or, more generally, unitaries. This approach corresponds to the initialization described in Section 3.1.1.

- **Canonical:** Each core is initialized randomly according to a Gaussian distribution, and the TT is subsequently brought into SVD-based canonical form.

*Remark* 5.1. Although the methods described in this work have primarily been used to tensorize NNs, they are equally applicable to other types of models. For example, instead of pre-training a complex model such as a NN, one could tensorize a simpler, easier-to-train model. While this approach may not yield the highest accuracies, it could suffice for initialization purposes, providing an appropriate output range that is replicated by the TT model. Alternatively, one could use as initialization for the TT model the tensorization of an initialized, un-trained NN.

The training accuracies of the TT models trained with the different configurations are shown in Fig. 8. While some combinations, such as using the polynomial embedding with the "Id. + noise" initialization or the unit embedding initialized with "Unitaries", appear to perform well during training, TT-RSS consistently yields the highest and most stable accuracies. However, this does not always translate into the best test accuracies. It is possible that other random initialization points allow for models that generalize better, whereas TT-RSS initialization may bias the training process of the obtained TT model.

Conversely, TT-RSS from random functions achieves significantly lower accuracies. Nevertheless, it is worth highlighting that this is an agnostic random initialization scheme—generalizable across different model configurations and independent of pre-trained models. Therefore, we consider the observed gradual improvement in accuracies a promising result. To further explore the potential of this initialization method, a more exhaustive search of training hyperparameters could be performed. Additionally, tensorizing alternative random functions that exhibit greater consistency across different inputs may enhance the method.

In both cases, "TT-RSS" and "TT-RSS random", tensorization enables the initialization of a TT model such that, for inputs drawn from the data distribution, it produces outputs within a controlled range that can be predetermined. This approach is also compatible with a large number of variables $n$, diverse embedding functions, and varying input/output or internal dimensions.

## 5.2 Compression

Tensor networks, particularly loop-less structures such as TTs and other tree-like networks, have been highly regarded in quantum information theory and condensed matter physics. Their popularity arises not only from the theoretical insights they enable but also from their computational efficiency, which makes it possible to simulate complex systems with relatively low computational effort. This efficiency has led to the adoption of TNs in various other fields to approximate complex computations effectively.

In deep learning, where models are composed of intricate combinations of linear layers followed by non-linear activations, TNs have typically been employed to reduce the number of parameters in the inner linear layers. Among the available tensor decomposition techniques, the most common approach involves converting each matrix in a linear layer into a Matrix Product Operator, also referred to as Tensor Train Matrix (TTM) [55, 58–62]. Other decomposition methods, such as the CP or Tucker decompositions, have also been successfully applied [56, 57]. This kind of tensorized models are commonly referred to as Tensor Neural Networks (TNNs).

Although this approach has shown promising results in compressing deep CNNs [55, 56, 61] and transformer-based language models [57, 60, 62], this type of compression retains two key drawbacks: (i) since the neural network architecture is preserved, the privacy and interpretability challenges of the original model remain present in the tensorized version, and (ii) although the decomposition significantly reduces the number of parameters that need to

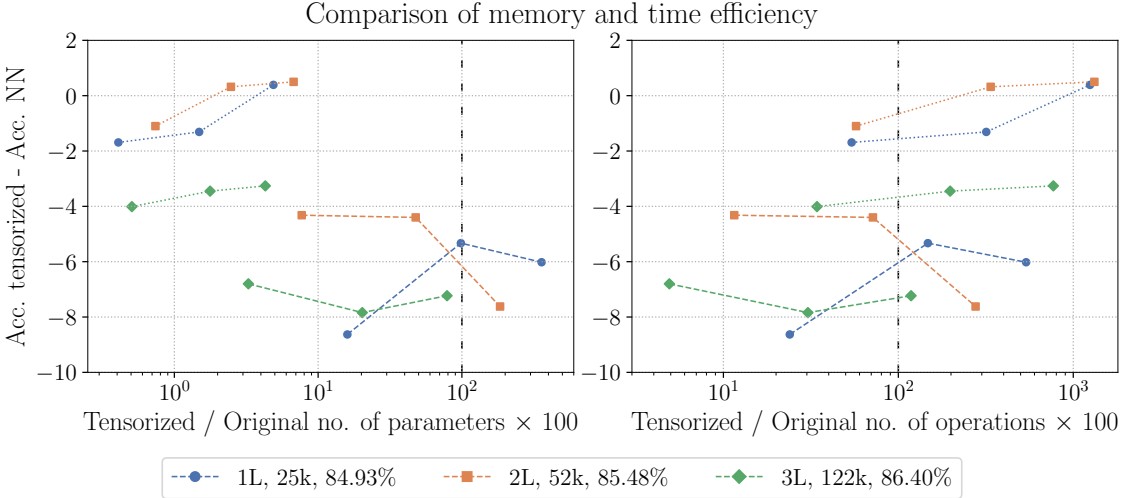

Figure 9: Comparison of memory and time efficiency between different tensorized models and various choices of ranks $r$. The x-axis represents the ratio of parameters/operations in the tensorized models to those in the original models. The y-axis marks the accuracy difference, where negative values indicate accuracy drops in the tensorized models compared to the originals. Each color corresponds to a different NN model being tensorized. The legend indicates the number of hidden layers, approximate number of parameters, and test accuracy of each model. Dashed lines represent TT models obtained via TT-RSS, while dotted lines denote layer-wise tensorized TNNs. Points connected by these lines indicate, from left to right, increasing ranks: $r = 2$, $r = 5$, and $r = 10$, with corresponding increases in the number of parameters and operations.

be stored—enabling edge computing on small devices—it does not always lead to a reduction in time complexity.

For example, in the base BERT model [5], some linear layers consist of matrices of size $768 \times 3072$. Both the storage and computational complexities of multiplying a vector by such a matrix scale with its dimensions. Specifically, storing the matrix requires $2\,359\,296$ floating-point numbers, and the multiplication involves $2\,359\,296$ floating-point operations.

On the other hand, if the same matrix were represented as a TTM, whose cores are of the form $G_k(\alpha_{k-1}, x_k, \alpha_k, y_k)$, where $y_k$ represents the output index of each core and all ranks are set to $r = 10$, we would need to store only $3\,850$ numbers. This represents a reduction of more than 99.8% in the number of parameters. However, contracting this TTM with an input vector (not in TT form) would require $2\,810\,880$ operations, which exceeds the computational cost of using the original matrix. This higher complexity arises even with a relatively small rank $r = 10$, which may already lead to significant approximation errors. The increased computational cost is primarily due to the need to recover the intermediate results as a full vector in order to apply non-linear activations element-wise, as these activations cannot be applied directly in TT form. The calculations assume the following decompositions for the input and output dimensions: $d^{in} = (1, 2, 2, 2, 2, 2, 2, 2, 2, 3, 1)$ and $d^{out} = (2, 2, 2, 2, 2, 2, 2, 2, 2, 2, 3)$.

To illustrate this phenomenon, we have tensorized three NN models with varying numbers of hidden layers, from 1 to 3, using both TT-RSS and layer-wise tensorization with TTMs. The models are trained for the binary classification task described in Section 4.1. For both tensorization methods, we consider three scenarios by setting $r = 2$, $r = 5$, and $r = 10$. In the case of layer-wise tensorization, this entails fixing the chosen rank for all layers. While this choice may not be optimal—since different layers have varying sizes, and a uniform rank may

affect them differently, thereby influencing the final accuracy—our primary goal is to compare the ability of these tensorization methods to reduce the number of parameters and computational costs. Nonetheless, to emphasize the expressive capacity of the resultant models, we re-train all tensorized models for 10 epochs, achieving accuracies comparable to those of the original NNs. The calculations focus solely on the linear layers of the models, assigning a constant cost to the application of the non-linear activations in the NN models.

The results are shown in Fig. 9. We observe similar results to those in the BERT-layer example discussed above. Although layer-wise tensorization can achieve orders-of-magnitude improvements in memory efficiency, it can also lead to orders-of-magnitude decreases in time efficiency. On the other hand, since it retains the same structure as the original model, it is capable of achieving similar accuracies. This demonstrates that layer-wise tensorization is a viable method for eliminating redundancies, thereby uncovering a finer internal structure within the model.

For TT-RSS, the expressive power of the resultant TT models may differ significantly from that of the original NNs, leading to larger accuracy drops. However, TT-RSS occasionally offers a better balance between memory and time efficiency, showcasing its potential as an alternative tensorization method for applications where such trade-offs are critical.

*Remark* 5.2. The calculations in this section assume the input vector is not in TT form. In practice, it is common to first represent the input vector as a TT, typically through iterative application of truncated SVD, which incurs a cost of $O(d^n r n)$. Subsequently, the TT input vector is contracted with the TTM layer, which can be performed efficiently with a cost of $O(d^2 r^4 n)$. However, the output vector still needs to be returned as a full vector to apply the element-wise non-linear activations, which requires $O(d^n r^2 n)$ operations. In contrast, contracting the full input vector with the TTM layer, as shown in our calculations, can be done in $O(d^{n+1} r^2 n)$. This demonstrates that both strategies have comparable complexities, and our results also extend to the former case, particularly in our scenario where $d = 2$.

# 6 Conclusions

In this work, we have presented a tensorization scheme that combines ideas from sketching and cross interpolation, resulting in an efficient method for function decomposition, TT-RSS. The key idea that allows to make the decomposition efficient is having black-box access to the function, together with a small set of points (the pivots) that define a subregion of interest within the function's domain. Typically, the number of pivots needed to cover the domain and obtain a faithful approximation of the function is exponential in the dimension. In TT-RSS, however, the function serves as an oracle, allowing for the interpolation of missing values between the pivots. This enables the recovery of a TT representation that accurately captures the behavior of the function at least within the chosen subregion, having only access to a polynomial amount of pivots.

While applicable to general functions, this method is particularly well-suited for machine learning models, where the subregion of interest is defined by the training dataset. Using only a small number of these samples, we demonstrate that it is possible to reconstruct models in TT form for tasks such as image and audio classification. This is especially relevant for NNs, where traditional tensorization techniques typically compress individual layers into TNs while preserving the network's overall structure and functionality.

To demonstrate the capabilities of the TT-RSS decomposition, we focus on scenarios where a TN representation is advantageous. The goal is not to train TT models from scratch, as this is readily achievable, but to explore cases where one might already have a trained NN model or prefer to train a NN for reasons of efficiency or expressiveness. In this context, we demonstrate

that TT-RSS can be used to obfuscate NN models, enhancing privacy protection through the well-characterized gauge freedom of TTs. Additionally, we have found that TT-RSS leads to representations that are interpretable. Concretely, it is possible to estimate physical quantities (in our example, the SPT phase of the AKLT state) from them.

While the AKLT example demonstrates the potential of the methodology we propose, the greatest advantage of our approach is likely to emerge in higher-dimensional scenarios. In such cases, training TNs becomes computationally expensive, whereas NNs can be trained more efficiently. Therefore, extending these methods to recover TN models for higher-dimensional layouts, such as 2D data, could offer a significant advantage over existing TN methods.

Moreover, TT-RSS can serve as an initialization technique for training TT models. One can first train a simpler model, such as a linear or logistic model, to obtain a function that returns values within a controlled range across the domain. Then, applying TT-RSS allows the construction of a TT model with the desired embedding, providing stable results in the domain of interest.

Finally, tensorizing NNs into a single TT model offers notable advantages in terms of reducing both the number of operations and the number of variables compared to the original network. This yields a better trade-off between memory and time complexity. In contrast, common tensorization techniques typically transform individual layers within a NN into TNs, reducing the number of variables but potentially increasing time complexity. Besides, these techniques do not address the privacy and interpretability concerns inherent in standard NN models.

In conclusion, the approach presented here provides a promising framework for efficiently decomposing NN models into TN forms, offering improvements in privacy protection, model interpretability, and computational efficiency. The potential for extending these methods to higher-dimensional problems is particularly compelling, as it could address the computational challenges inherent in training TNs in such settings while retaining their advantages over standard NN models.

## Acknowledgments

**Funding information**  This work is supported by the Spanish Ministry of Science and Innovation MCIN/AEI/10.13039/5011 00011033 (CEX2023-001347-S, CEX2019-000904-S, CEX2019-000904-S-20-4, PID2020-1135 23GB-I00, PID2023-146758NB-I00), Comunidad de Madrid (QUITEMAD-CM P2018/TCS-4342, TEC-2024/ COM-84-QUITEMAD-CM), Universidad Complutense de Madrid (FEI-EU-22-06), the CSIC Quantum Technologies Platform PTI-001, the Swiss National Science Foundation (grant number 224561), and the Ministry for Digital Transformation and of Civil Service of the Spanish Government through the QUANTUM ENIA project call - Quantum Spain project, and by the European Union through the Recovery, Transformation and Resilience Plan - NextGenerationEU within the framework of the Digital Spain 2026 Agenda. This research was supported in part by Perimeter Institute for Theoretical Physics. Research at Perimeter Institute is supported by the Government of Canada through the Department of Innovation, Science, and Economic Development, and by the Province of Ontario through the Ministry of Colleges and Universities.

**Code availability**  The source code for the experiments is hosted on GitHub and publicly available at: https://github.com/joserapa98/tensorization-nns.

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
