# Peer review of "Tensorization of neural networks for improved privacy and interpretability"

_SciPost Physics Core, doi:SciPost Phys. Core 8, 095 (2025)_

## Round 1 · Referee Report · Anonymous (Referee 1) · 2025-3-6

Strengths

  1. Introduce a practical heuristic algorithm for determining a TT approximation from sampled values
  2. Demonstration in a various range of tasks with a special focus on privacy and interpretability

Weaknesses

  1. It is not written what are the advantages of TT-RSS over TT-RS
  2. No direct comparisons of efficiency with similar technologies [Tensor Train Cross Interpolation (TT-CI)]

Report

This manuscript modifies TT-RS (TT recursive sketching) and numerically demonstrates the algorithm’s efficiency. The authors also apply the method to improve the privacy and interpretability of tensor trains.

The proposed algorithm constructs a TT representation from the values of a function at selected sampling points. As in previous studies, the algorithm uses the Core Determining Equations (CDEs) to determine the TT cores, but it employs sketches and random projections to make solving the CDEs feasible. Overall, the algorithm resembles TT-RS (Ref. 63), with the main differences being the choice of sketching matrices and the random projections used.

The manuscript is accessible to readers who are not already familiar with TT-RS, and its preliminary applications to machine learning will likely attract interest from researchers outside of physics. I believe the manuscript meets the criterion of “providing a novel and synergetic link between different research areas” along with other general acceptance criteria. However, I would like to hear reviews from applied mathematicians and machine learning experts as well, if possible.

I recommend that the authors make some modifications to clarify what is new in their approach. Below are my specific comments:

1. In Ref. 63, Hur et al. proved conditions for suitable sketch matrices (Theorem 3). Does the choice of sketch matrices in the present study satisfy these conditions?
2. The authors state that TT-CI does not apply to continuous functions. However, Ref. 77 (Appendix A.1) presents a TT-CI algorithm for continuous functions. Please clarify this discrepancy.
3. I did not understand the point of the sentence: “Although we do not formally establish properties that the selected pivots must satisfy to yield good results, our proposal of choosing pivots from the training set…” Is the intended meaning that, when approximating a neural network model trained on a dataset, choosing pivots randomly from that same training set is empirically effective? If so, what is a robust method for selecting good pivots from a black-box function? Random sampling in an exponentially large space is likely to suffer from the curse of dimensionality.
4. In Equation (2.10), the authors introduce random square projection matrices. The role of these projections is unclear. They appear to mix the columns of bar(Phi) in Equation (2.13), yet after the singular value decomposition in Equation (2.14), the right matrix, Ck, is discarded. Please clarify the purpose and impact of this projection on the overall algorithm.

Requested changes

1. Section 2 is very similar to the description in Ref. 63. To clarify the differences and fill in some gaps in the logic, please refer to Ref. 63 more frequently. For example, the introduction of the A matrix in Equation (2.15) appears rather abrupt without additional context.

2. The statement, “Moreover, Ref. [73] demonstrated that Equation (2.29) yields a TT representation of f if the interpolation sets …” is misleading. When pivots are nested, the TT-CI formula interpolates the pivots. If not nested, the TT-CI formula only approximates the function.

Recommendation

Ask for minor revision

  • validity: ok
  • significance: ok
  • originality: ok
  • clarity: good
  • formatting: good
  • grammar: good

Author:  José Ramón Pareja Monturiol  on 2025-08-07  [id 5707]

(in reply to Report 1 on 2025-03-06)

We thank the referee for their time dedicated to review our manuscript, their positive feedback and constructive suggestions. In response to the concerns raised, we have revised the manuscript to clarify the distinctions between our method and prior works, both TT-RS and TT-CI. Specifically:

  • We added clarifying remarks and citations throughout Section 2 (lines 409–425, 441–442, 449–451, 458–461, 592–592, 595–599, 609–617).

  • We included new subsections 3.1.3 and 3.2.3, along with Tables 3.1 and 3.2, to directly compare the performance of TT-RSS and TT-CI on the examples discussed in Sections 3.1 and 3.2.

  • All experiments in these sections, as well as those in Section 4.2, were recomputed using double precision. The results remain largely unchanged, with the main difference being that the errors for exactly recoverable random TTs now reach the updated machine precision. Time costs for TT-RSS increased due to the higher precision. Since no accuracy improvements were observed for models without exact TT structure, we did not recompute other experiments.

We address the referee’s specific comments below.

Weaknesses

  1. “It is not written what are the advantages of TT-RSS over TT-RS”: We thank the referee for bringing up this point. Ref. [63] presents a general framework using random sketches. This framework is, however, applicable efficiently only in two settings: (i) empirical distributions (in low dimensions), and (ii) Markov chains, for which specific sketches are designed to yield exact recovery. These are not the types of problems we aim to address. Our goal is to tensorize machine learning models used for image and audio classification tasks—settings characterized by high dimensionality and sparsity. As explained in Remark 2.6, applying the TT-RS scheme with only a polynomial number of training samples to build an empirical distribution leads to random TTs. Therefore, the main difference of our approach with TT-RS is that, instead of tensorizing empirical distributions, we tensorize the models themselves, which can generalize and fill the gaps between the limited number of available samples. To maintain computational efficiency, we replace random sketches—which scale exponentially with input dimension as rightfully identified by the referee—with “cross interpolation” sketches. These are based on subsets of rows and columns determined by selected pivots. In the new version of our manuscript that we are hereby resubmitting we have clarified the advantages of our approach with respect to prior work in lines 409–425, 441–442, and 458–461. In addition to this, in Section 2.5 (lines 592–599 and 609–617), we now include a more detailed comparison with TT-CI, highlighting that identifying a good set of pivots is essential. In our problem setting, the maxvol routine used in TT-CI is not effective at adequately exploring the domain.

  2. “No direct comparisons of efficiency with similar technologies [Tensor Train Cross Interpolation (TT-CI)]”: We completely agree with the referee in that a comparison with alternative existing methods will help understanding the scope and actual impact of our results. We now include such comparisons in two new sections, 3.1.3 and 3.2.3. The summary of the results in these new sections is that TT-CI performs well in simple cases but incurs higher computational cost. In more complex tasks like MNIST classification, TT-CI fails to yield accurate approximations, whereas TT-RSS performs significantly better, producing TT models that achieve around 80% accuracy from a direct tensorization of the full neural network classifiers.

Comments

  1. Indeed, Ref. [63] proves that if the sketched matrices Φk​ preserve the row and column spaces of the original matrices, the CDEs admit unique, exact solutions. While this is straightforward conceptually, constructing such sketches is non-trivial. Ref. [63] only provides explicit constructions for Markov chains. Using random sketches offers probabilistic guarantees (via random matrix theory), but at exponential cost. Our work introduces a practical strategy: construct sketches from domain or training set samples using cross interpolation principles, and optionally apply random projections Uk​ to reduce variance. While we agree that having a guarantee of optimality would be desired, in this work we focus on empirical performance, leaving formal proofs for future work. Importantly, our goal is approximation of the underlying density, and exact TT reconstruction of the model is not required.

  2. We have noticed that our choice of wording when discussing extensions to continuous functions in lines 216–217 may have conveyed an incorrect message. Throughout the manuscript (see lines 183–186 and 230–231) we explicitly write that TT-CI can be applied to continuous functions. However, this typically involves discretization, which increases the number of input variables and can lead to large, high-rank TTs. For models with hundreds of features, this is inefficient and contradicts our compression goals. In contrast, Ref. [63] applies to continuous functions using continuous feature embeddings, avoiding the dimensionality blow-up. While such embeddings could theoretically be integrated into TT-CI, to our knowledge, this has not yet been done. In the resubmitted version we are re-phrasing lines 216–217 to avoid potential misunderstandings in this regard.

  3. Indeed, the meaning inferred by the referee is the intended one. In other words, in our work we do not provide formal guarantees on pivot selection but follow heuristics aligned with TT-CI and the max-volume principle. Empirically, for ML models, using training set points as pivots works well and often outperforms TT-CI in less time. For smoother functions, uniformly random samples from the domain suffice. Importantly, as noted in lines 1363–1367, although we use a polynomial number of pivots, we evaluate the function on their Cartesian product to sketch the tensors, effectively covering a larger region of the domain. Moreover, recovering a TT of rank r only requires r samples per step (typically oversampled to reduce error), so our method avoids the exponential cost correctly described by the referee. In the resubmitted version, we elaborate on this fact in lines 409–425. In addition to this, we agree with the referee in that finding provably robust strategies for choosing the pivots is a very important and natural follow-up question to our work. We now point explicitly to this open question in line 581 in the resubmitted manuscript.

  4. The random projection matrices Uk follow the logic of random sketches from Ref. [63], enabling recovery of row and column spaces. In our case, they are applied after the orthogonal projections P, which are introduced to improve efficiency. In the resubmitted version of our manuscript, we explain in Remark 2.3 (lines 449–451) that these projections can act as Johnson-Lindenstrauss embeddings, preserving pair-wise distances while reducing variance, and potentially reducing SVD costs in the trimming step. In our examples, we use square projections Uk​, since the matrix sizes are already manageable and the SVD alone would be more effective in capturing relevant subspaces.

Requested changes

  1. As correctly noted by the referee, Section 2 indeed mirrors the structure of Ref. [63], as we acknowledge in lines 367, 381, 427, 471, and 511 of the manuscript. To further emphasize our contributions, in the resubmitted version we have added references and clarifications in lines 441–442 and 458–461. In the particular case of the matrices Ak​ in Section 2.2.4, we believe that no further referencing is needed, since it is simply a definition, and all of its components have already been introduced earlier in the text.

  2. We agree with the referee in that the sentence mentioned was confusing. In the resubmitted version of the manuscript, it has been revised (lines 592–593 and 595–599) to clarify that TT-CI yields exact interpolation only with nested pivots. Since we apply SVD, exact interpolation is not achieved in our method by design, but this is intended in order to avoid overfitting.

---

## Round 1 · Referee Report · Anonymous (Referee 2) · 2025-6-24

Strengths

  1. The problem of fitting data to a matrix product state is of clear interest and timely.

Weaknesses

  1. This is not a physics paper.
  2. The performances of the approach are unclear. The algorithm is insufficiently characterised. (At first glance the performances look inferior to existing algorithms).
  3. No comparison with other approaches has been performed.

Report

The article describes an extension of a sketching algorithm to fit a function to a matrix product state.
The resulting RSS algorithm is applied to the compression of neural networks and in particular to the problem of privacy (making the training data unretrievable from the neural network) and interpretability.

I had a hard time to figure out if the method was working or not. For instance, I assume that Fig.3.1 plots R(s) and that the left panel is multiplied by 10^-6. But what about the middle panels? If it is also multiplied by 10^-6 then the method works fine, I am just surprised that it works better on the test than on the training data. But I am puzzled by the statement in the “slater” subsection: “Although these results are less conclusive than those of the previous case, the bottom row of Fig. 3.1
shows that the errors decrease as N increases. Remarkably, even with a higher discretization level of
l=100, we achieve test errors on the order of 10^-2 for N> 70” which does not match with the above interpretation (10^-2 >> 10^-6). In any case the first example being exactly of finite rank, the error should drop to machine precision and it clearly does not. There is something I miss here. Also, how does N translates in terms of the number of function calls is unclear.
The second “slater” example is the function exp(-r) with r=\sqrt{x^2+y^2+z^2+…}. I may be mistaken but my feeling is that a technique like Tensor Cross Interpolation would give much better results than a mere two digits. Overall, the authors should clarify the status of these results.

I think that the application to interpretability and privacy does not make much sense if the results are poor. I had an even harder time interpreting these results but e.g. the log scale on the lower panel of Fig.3.2 test error column looks pretty bad to me. Same with later figures, accuracies at the 50% level are not really useful. The authors argue that this may be a baseline improved later for “privacy” purpose. Perhaps, but I am not fully convinced. The AKLT example is not fully convincing too: it can be recovered easily by other methods.

So overall, I found that the article lacks in clarity as far as the results are concerned, and I am afraid that this lack of clarity simply hides the fact that the results are not so good.

Requested changes

  1. Compare the performance to other algorithms such as Tensor Cross Interpolation.
  2. Clarify the performances of RSS. Include less trivial examples.
  3. Some figures (e.g. 3.1, 3.2… ) have no clear X-axis and Y-axis. Please update.
  4. No study of the behaviour of the approximation versus c is shown so we don’t know the limits of the approach. Please update.
  5. Same with the number of function calls

Recommendation

Ask for major revision

  • validity: low
  • significance: ok
  • originality: good
  • clarity: ok
  • formatting: good
  • grammar: good

Author:  José Ramón Pareja Monturiol  on 2025-08-07  [id 5708]

(in reply to Report 2 on 2025-06-24)

We thank the referee for their time and effort dedicated to reviewing our manuscript. We appreciate their recognition of our work as addressing a problem “of interest and timely”, yet we respectfully but firmly disagree with their assessment that “[a perceived lack of clarity from the referee’s point of view] simply hides the fact that the results are not so good”. We want to stress that our choice of experiments is always exclusively motivated by the aim to provide the clearest and fairest assessment of the technique. Still, we acknowledge the fact that some parts of the manuscript can be written in a clearer way, and thus we are taking this chance for resubmission as an opportunity to clarify the specific concerns raised by the referee. Concretely, regarding the concerns raised in the referee’s report, in the resubmitted manuscript:

  • We added clarifying remarks throughout the manuscript (lines 409–425, 441–442, 449–451, 458–461, 592–592, 595–599, 609–617).

  • We included new subsections 3.1.3 and 3.2.3, along with Tables 3.1 and 3.2, to directly compare the performance of TT-RSS and TT-CI on the examples discussed in Sections 3.1 and 3.2.

  • In the tensorization of random TT functions, we increased the number of pivots considered from the range 10–20 to 10–35 to better illustrate how the test error decreases as the number of pivots increases.

  • All experiments in these sections, as well as those in Section 4.2, were recomputed using double precision. The results remain largely unchanged, with the main difference being that the errors for exactly recoverable random TTs now reach the updated machine precision. Time costs for TT-RSS increased due to the higher precision. Since no accuracy improvements were observed for models without exact TT structure, we did not recompute other experiments.

  • We updated figures and their captions in order to depict our results more clearly and avoid potential misinterpretations.

In the following, we address the referee’s specific comments:

Weaknesses

The referee states as a weakness of our work that it “is not a physics paper”. While we respect their opinion in this matter and we do not contend the fact that one of the main application of our method is machine learning, we want to note here that: (i) the tools we employ, namely tensor networks, have a long and rich history in quantum many-body physics, and (ii) another of the main applications that we describe in our manuscript, and that we illustrate with simple examples, is the extraction of insights of physical relevance from neural-network quantum states. Moreover, we believe that here it may be relevant to quote Referee #1’s report, which reads “[this work] meets the criterion of providing a novel and synergetic link between different research areas". We believe that it is clear that the areas that Referee #1 refers to in this excerpt are machine learning and, precisely, physics.

The remaining weaknesses refer to the performance of the techniques developed and their comparison with other existing techniques. In order to address them, in the version of the manuscript that we are hereby resubmitting we have added a direct comparison with TT-CI (Tables 3.1 and 3.2) and clarified the distinctions between our method and TT-RS/TT-CI throughout the manuscript (see lines 409–425, 441–442, 449–451, 458–461, 592–592, 595–599, 609–617). These changes aim to better contextualize the performance of TT-RSS and explain why TT-RS is not feasible for our setting.

Comments

  • “For instance, I assume that Fig.3.1 plots R(s) and that the left panel is multiplied by 10^-6. But what about the middle panels? If it is also multiplied by 10^-6 then the method works fine, I am just surprised that it works better on the test than on the training data. But I am puzzled by the statement in the “slater” subsection: “Although these results are less conclusive than those of the previous case, the bottom row of Fig. 3.1 shows that the errors decrease as N increases. Remarkably, even with a higher discretization level of l=100, we achieve test errors on the order of 10^-2 for N > 70” which does not match with the above interpretation (10^-2 >> 10^-6)”: Figure 3.1 plots different quantities in the different columns. As explained in both the labels for the columns and the corresponding caption, the left panels plot the relative error on the pivots, R(x); the middle panels plot the relative error on a test set, R(s); and the right panels plot the time in seconds needed to perform the tensorization. Importantly, the only plot where the vertical axis is multiplied by 10^-6 is the top-left one, corresponding to the error on pivots for tensorizations of random TTs. In the updated figure, the order of magnitude on the vertical axis is 10^-14, instead of the original 10^-6. As we discuss below in the response, this is due to the fact that we have re-performed the computations at double precision, so that we reach machine precision when tensorizing functions that admit a tensor network form. For functions that do not admit exact TT representations, such as the Slater functions, the magnitude of the errors one should expect is unclear, and depends on the expressivity of the TT model (specified by the chosen ranks). In Table 3.1, which we have included in the resubmitted version of the manuscript, we show that, indeed, the resulting errors for Slater functions are comparable to, and often better than, those obtained with TT-CI, supporting the validity of our results. We hope that the new Tables 3.1 and 3.2 also clarify the concern of the referee that was pointed out in this comment—namely, that the way in which results were reported in Figures 3.1 and 3.2 might be a possible source of confusion. We would like to note that, due to the randomness in TT-RSS, some applications of the algorithm may yield high errors, resulting in high variance and deviations of the mean values. Therefore, although Figures 3.1 and 3.2 are useful to illustrate general trends, reporting only means and standard deviations does not reliably capture typical behavior. To better reflect typical performance, Tables 3.1 and 3.2 report median values for both errors and runtimes. As seen there, test errors can reach machine precision in the median, even with small pivot sets (e.g., N = 20).

  • “In any case the first example being exactly of finite rank, the error should drop to machine precision and it clearly does not”: The fact that the errors reach only magnitudes of the order of 10^-6 is motivated by the fact that computations were performed in single precision. Taking this fact into account, the results are compatible with perfect recovery. In the resubmitted version, we are replacing the corresponding plots with their counterparts computed in double precision, showing that now the errors are of the order of 10^-15, i.e., close to machine precision. In addition to this, all the relevant captions (in Figures 3.1, 3.2, and 4.4, and in the new Tables 3.1 and 3.2) stress the fact that computations are performed in double precision.

  • “Also, how does N translates in terms of the number of function calls is unclear”: Our sketches follow a cross interpolation scheme, requiring function evaluations on the Cartesian product of the sets x_{1:k-1}, [d_k], and x_{k+1:n}, as detailed in Eq. 2.12, in the surrounding discussion, and in Remark 2.4. This procedure is what determines the number of function calls. As noted in Remark 2.2, we eliminate duplicates when forming subsets x_{a:b}​, so the actual number of unique evaluation points N_{a:b}​ can be smaller than N. As a result, the precise count of function calls depends on the specific pivot set, but in general it scales as O(N^2 d). In the resubmitted manuscript, we have modified lines 458–459 and 557 to further clarify this aspect. We note that since the duplicate removal is minor, we report performance with respect to the full pivot count N, rather than the exact number of function calls.

  • “I may be mistaken but my feeling is that a technique like Tensor Cross Interpolation would give much better results than a mere two digits. Overall, the authors should clarify the status of these results”: In the resubmitted manuscript we have included Tables 3.1 and 3.2, which provide a direct comparison between TT-CI and TT-RSS across all experiments, with Sections 3.1.3 and 3.2.3 detailing the procedure used to apply TT-CI and the faithfulness of our results. Overall, TT-RSS achieves performance comparable to TT-CI in function reconstruction and toy machine learning problems, while significantly outperforming it in more complex machine learning scenarios, which are our main focus. Additionally, in all cases, TT-CI requires substantially longer computing times—typically one order of magnitude larger than those for TT-RSS. For the Slater functions, as noted earlier in our response, we compare our method with TT-CI in Table 3.1 and report similar error levels, confirming that Tensor Cross Interpolation also yields only one or two digits of precision. The motivation for selecting these functions stems from their use in a previous work, Ref. [110], as noted in line 689. In that work, Table 1 presents decomposition results for Slater functions with parameters L=10, m=3, and l between 10 and 20, with maximum TT ranks reaching 7, 16, or 32. They report accuracies on the order of 10^-3 to 10^-5. However, we emphasize three key differences: (i) their reported accuracy refers to the error of each individual cross interpolation decomposition, not to test error of the full TT approximation, which likely scales with the number of tensors in the TT chain (20–60 in their case); (ii) their problems are simpler, with lower dimensionality (m=3) and coarser discretizations (l=10–20), whereas we consider m=5 and l=30–100; and (iii) they allow higher ranks (up to 32), while we fix the maximum rank to 10, reducing expressivity. For these reasons, we believe our method achieves state-of-the-art performance, with errors in the range of 10^-1 to 10^-2.

  • “I think that the application to interpretability and privacy does not make much sense if the results are poor”: In the new Section 3.2.3 (lines 863–870), we explain in more detail that the accuracy drop is more pronounced in MNIST, likely due to its 2D data layout, which does not align well with the 1D structure of TT models. In contrast, in the privacy experiments with audio classification models, the data better matches the TT structure, resulting in smaller accuracy drops. As shown in Figure 4.1 and lines 1006–1011, the accuracy drop in the privacy experiments is around 10% directly after tensorization, and only about 3% after re-training for just 10 epochs. This results in models with accuracies comparable to the original neural network classifiers, but with much stronger privacy guarantees. As shown in Figure 4.3, the tensorized models are robust against our privacy attacks. Specifically, under attacks where the adversary is assumed to know the model architecture, has access to its parameters, and enough computational power to train multiple models under different assumptions, the attacker’s prediction based on TT model parameters is essentially random, with a 50% success rate (Figure 4.3, right panel). To our knowledge, this is the first time a method like ours has been applied successfully to real-world scenarios. Protecting models in this way comes at the cost of only a 3% accuracy loss, which is acceptable given that balancing accuracy and privacy is fundamental in privacy-preserving methods such as Differential Privacy, where stronger privacy always reduces accuracy. Regarding the interpretability experiments (discussed below), our goal is to demonstrate a methodology applicable even when it is not clear how to obtain a tensor network representation of the target state. We use the AKLT state as a case study, where the expected result is known. Remarkably, we are able to reconstruct the exact state using only about 10 random spin configurations as pivots, for chain lengths up to 500, and in under 20 seconds.

  • “the log scale on the lower panel of Fig.3.2 test error column looks pretty bad to me”: Indeed, the results on the lower panel of the test error column in Fig. 3.2 may seem a priori surprising, especially when compared with those in the corresponding upper panel. Its explanation appears in lines 817–822 in the manuscript. In short, the tensorization may fail to recover the underlying function in non-pivot points, but it nevertheless still acts as a good machine learning model (where only coarse-grained information such as the sign of the function is used as the class prediction), as demonstrated in the third column of the figure when the number of pivots is increased. Still, we agree that this fact may not be obvious and the manuscript could benefit from extended explanations. For this reason, in the resubmitted version, we describe again this fact when discussing the results in the new Section 3.2.3 (lines 851–862).

  • “Same with later figures, accuracies at the 50% level are not really useful“: We assume here that the referee is referring to the results shown in the third column of Fig. 3.2. In this figure, we report the percentage of datapoints in the test set for which the predicted label differs between the original model and its tensorization. We believe this is a metric that is both easy to compute and effective at capturing the differences between the original and tensorized machine learning models—unlike the first and second columns, which are better suited to analyzing their differences as continuous functions. A value of 50% in these plots means that the tensorized model assigns different labels from the original model to half of the test set. The purpose of these figures is to illustrate the trend as the number of pivots, N, increases. Indeed, as N increases, the percentage of differing label assignments decreases, reflecting the fact that the tensorized model becomes increasingly similar to the original.

  • “The authors argue that this may be a baseline improved later for “privacy” purpose”: As mentioned earlier, to reduce the accuracy drop in the privacy experiments caused by the tensorization of neural networks, we further optimize the tensorized models via gradient descent to recover accuracies closer to the original ones. This results in a final degradation of just 3% in accuracy. More generally, although tensorization typically affects accuracy, the tensorized models can always be further optimized, benefiting from a good and stable initialization. This motivates Section 5.1, where we argue that tensorizing models can serve as a general initialization strategy for TT models.

  • “The AKLT example is not fully convincing too: it can be recovered easily by other methods”: We agree with the referee in that the AKLT model is a paradigmatic one in condensed matter theory, and as such has been thoroughly studied and there exist many ways of obtaining its ground state. In Section 4.2, specifically in lines 1202–1212, we explained that the AKLT example serves as a proof of principle for a method to obtain physical insights from black-box access. The actual goal, that is only illustrated with the AKLT example, is to propose a methodology agnostic to prior knowledge of the system's representation. We argue that this approach could be especially useful if the TT-RSS decomposition or its underlying ideas can be extended to decompose neural network quantum states into other tensor networks, particularly for 2D systems.

Requested changes

  1. This comparison is now included in subsections 3.1.3 and 3.2.3 of the resubmitted manuscript.

  2. We sincerely hope that the new discussions, theoretical complexity analyses, and experiments added in response to the comments from both referees—described above—help clarify the performance of TT-RSS. Regarding the request for additional examples, we believe that the current set (tensorizations of exact TTs, smooth functions, toy machine learning models such as Bars and Stripes, models trained on benchmark datasets like MNIST, models trained on complex datasets such as CommonVoice, and quantum many-body states accessed via black-box evaluation) is sufficient to faithfully illustrate the potential and limitations of the technique.

  3. We have clarified in the captions of Figures 3.1 and 3.2 that the x-axis represents the number of pivots N.

  4. To measure decomposition performance precisely, we tune parameters that explicitly and controllably affect the resulting model. The TT model’s expressivity is constrained by input dimension d and ranks r, while the number of pivots N strongly correlates with the achievable approximation error. Parameter c measures the relative error allowed for each SVD in TT-RSS; although related to the ranks, specifying c alone is imprecise because it can modify the ranks of the tensors in an uncontrollable manner. Typically, we fix an upper bound for ranks r (commonly saturated) and use a very small c to allow flexibility and adaptively find smaller ranks by removing smallest singular values. Therefore, while c is useful as a measure of relative error, it does not controllably relate to parameters that characterize the final model. We believe it is more precise to measure performance by the maximum rank r, as we do in Figure 3.3.

  5. The revised version of the manuscript includes further discussion of this aspect in Sections 2.2 and 2.4. Specifically, we state explicitly that the number of pivots, N, determines the number of function calls as O(N^2dn). Consequently, all results in Section 3 already analyze the behavior of the methods as a function of the number of calls. In particular, Figures 3.1 and 3.2 show that the time costs scale polynomially with N.

---

## Round 2 · Referee Report · Anonymous (Referee 3) · 2025-10-16

Report

I believe that Referee 2 has a valid point: This is not a physics paper (see item 1 in "Weaknesses" of Report #2 by Referee 2 on 2025-6-24). For example, the MNIST data set discussed in section 3.2.2 definitely is not physics, neither are many of the other points discussed in this manuscript.

Furthermore, I have an issue with the context of the one point that actually is physics, namely the AKLT model discussed in section 4.2.1. Between Eqs. (4.6) and (4.7), the authors refer to their own relatively recent 2021 review [33]. However, if one looks at appendix 1.e of Ref. [33], this refers back to the original AKLT paper Phys. Rev. Lett. 59, 799 (1987). The notation used in section 4.2.1 of the present work might not be obvious from the original AKLT reference, but I understand that this $2 \times 2$ matrix-product state representation is known since at least 30 years. Take for example the review article by Hans-Jürgen Mikeska and Alexei K. Kolezhuk, Lecture Notes in Physics 645, 1 (2004), https://doi.org/10.1007/BFb0119591. There, in section 1.3.3, one not only finds a very similar representation with $2 \times 2$ $g$ matrices, but also references to early publications on the matrix-product state interpretation of the AKLT and related states such as M. Fannes, B. Nachtergaele, R. F. Werner, Europhys. Lett. 10, 633 (1989); Commun. Math. Phys. 144, 443 (1992); A. Klümper, A. Schadschneider, J. Zittartz, J. Phys. A 24. L955 (1991); Z. Phys. B 87, 281 (1992); Europhys. Lett. 24, 293 (1993).

I believe that this context issue needs to be fixed, i.e., the authors need to make it clear that the AKLT story is much older than the current presentation suggests. Once this is done, I believe that the manuscript can be published in SciPost Physics Core.

Requested changes

Place the AKLT story properly into its historical context.

Recommendation

Accept in alternative Journal (see Report)

  • validity: -
  • significance: -
  • originality: -
  • clarity: -
  • formatting: -
  • grammar: -

Author:  José Ramón Pareja Monturiol  on 2025-11-16  [id 6039]

(in reply to Report 1 on 2025-10-16)
Category:
answer to question

We thank the referee for the constructive comment and for pointing out the need to better contextualize the AKLT model within its historical development. Following this suggestion, we have added a short clarification in Section 4.2.1 (lines 1097–1098) explicitly acknowledging that the TT/MPS representation of the AKLT state was derived in early works, and we now cite the relevant foundational references [Fannes, Nachtergaele, and Werner, Commun. Math. Phys. 144, 443–490 (1992); Klümper, Schadschneider, and Zittartz, Z. Phys. B 87, 281–287 (1992)].

We believe this addition properly places our presentation in its historical context and addresses the referee’s concern.

---

## Round 2 · Author Response

We thank the Editor and the Referees for their time and effort dedicated to reviewing our manuscript. We acknowledge that assessing interdisciplinary works is oftentimes challenging, and we recognize the efforts made to provide a fair refereeing process. We have revised the text to address all the comments and suggestions raised during the review process.

In response to the detailed feedback from the referees, we have submitted a revised version of our manuscript that addresses all the concerns raised. We have made a particular effort to clarify the distinctions between our method and previous works, especially TT-RS and TT-CI, which was a concern shared by both referees. To this end, we have included clarifying remarks and citations throughout Section 2 and introduced two new subsections (3.1.3 and 3.2.3), along with Tables 3.1 and 3.2, which present direct performance comparisons between TT-RSS and TT-CI on representative examples. Additionally, all experiments in Sections 3.1, 3.2, and 4.2 have been recomputed using double precision, and as a result, we have updated Figures 3.1, 3.2, and 4.4. While the results remain largely consistent with those originally reported, we hope that the updated figures, together with the new tables, help clarify the potential of our approach.

We hope that these changes address the reviewers’ concerns and that our manuscript is now suitable for publication in SciPost Physics.

---

## Round 2 · List of Changes

• We added clarifying remarks and citations to Section 2 to better distinguish TT-RSS from prior works (TT-RS and TT-CI), addressing a shared concern by both referees. These edits span lines 409–425, 441–442, 449–451, 458–461, 592–592, 595–599, and 609–617.
  • We introduced two new subsections 3.1.3 and 3.2.3 along with Tables 3.1 and 3.2. These present direct numerical comparisons between TT-RSS and TT-CI on the representative examples discussed earlier in the section.
  • All experiments in Sections 3.1, 3.2, and 4.2 have been recomputed using double precision, and as a result, we have updated Figures 3.1, 3.2, and 4.4.
  • We updated the captions of Figures 3.1 and 3.2 to clarify that the x-axis represents the number of pivots N. In addition, we revised the captions of these figures, as well as that of Figure 4.4, to indicate that double precision is used only in those computations.
  • In the tensorization of random TT functions, we increased the number of pivots considered from the range 10–20 to 10–35 to better illustrate how the test error decreases as the number of pivots increases.
  • Minor corrections and clarifications were made throughout the manuscript for improved readability and consistency.

---

## Round 3 · Author Response

We hope this clarification resolves the remaining concern and that the manuscript is now suitable for publication in SciPost Physics Core.

---

## Round 3 · List of Changes



---

## Editorial Decision

published